# A 28 time-points cropland area change dataset in Northeast China from 1000 to 2020

Ran Jia[1,3], Xiuqi Fang[1,2], Yundi Yang[1], Masayuki Yokozawa[3], Yu Ye[1,2]

[1]Faculty of Geographical Science, Beijing Normal University, Beijing 100875, China

[2]Key Laboratory of Environmental Change and Natural Disaster, Ministry of Education, Beijing Normal University, Beijing 100875, China

[3]Faculty of Human Sciences, Waseda University, 2-579-15 Mikajima, Tokorozawa 359-1192, Japan

*Correspondence to*: Yu Ye (yeyuleaffish@bnu.edu.cn)

**Abstract.** Based on historical documents, population data, published results, remote sensing data products, statistical data and survey data, this study reconstructed the cropland area and the spatial pattern changes at 28 time points from 1000 to 2020 in Northeast China. 1000 to 1600 corresponds to historical provincial-level administrative districts, while 1700 to 2020 corresponds to modern county-level administrative districts. The main findings are as follows: (1) The cropland in Northeast China exhibited phase changes of expansion-reduction-expansion over the past millennium. (2) the cropland area in Northeast China increased from $0.55 \times 10^4$ km$^2$ in 1000 to $37.90 \times 10^4$ km$^2$ in 2020 and the average cropland fraction increased from 0.37% to 26.27%; (3) from 1000 to 1200, the cropland area exhibited an increasing trend, peaking in 1200. The scope of land reclamation was comparable to modern times, but the overall cropland fraction remained low. The cropland area significantly decreased between 1300 and 1600, with the main land reclamation area was reduced southward into Liaoning Province. From 1700 to 1850, the cropland area increased slowly, and the agricultural reclamation gradually expanded northward. After 1850, there was almost exponential growth, with the cropland area continuously expanding to the whole study area, and the growth trend persists until 2020; (4) the dataset of changes in cropland of administrative districts in Northeast China, reconstructed based on multiple data sources and improved historical cropland reconstruction methods, significantly enhances time resolution and reliability. Additionally, the dataset shows relatively better credibility assessment results, which can provide a refined data base for historical LUCC dataset reconstruction, carbon emission estimation, climate data construction, etc. The dataset can be downloaded from https://doi.org/10.6084/m9.figshare.25450468.v2 (Jia, 2024).

## 1 Introduction

Anthropogenic land cover change (ALCC) is a key driver of global change, significantly impacting climate change (Arneth et al., 2017; Foley et al., 2005; Ito and Hajima, 2020; Ellis et al., 2021; Roberts, 2019), over 70% of the Earth's land surface has undergone anthropogenic alterations over the past millennium (Sebastiaan et al., 2014; Shukla et al., 2019; Winkler et al., 2021). Cropland constitutes one of the primary land use types, being a land category susceptible to human influence and undergoing alterations, and it significantly influences food security, soil health, biodiversity, greenhouse gas emissions, and

climate change (Friedlingstein et al., 2023; Godfray et al., 2010; Kalnay and Cai, 2003; Poschlod et al., 2005). Additionally, in recent years, croplands cover 12~14% of the global ice-free land (Shukla et al., 2019), Research on the long-term, accurate temporal and spatial changes in cropland are crucial for understanding the carbon budget resulting from human land reclamation, tracking sustainable food production, and other land-based ecosystem functions (Huang et al., 2024; Potapov et al., 2022; Saez-Sandino et al., 2024; Yu and Lu, 2018).

Presently, various global historical Land Use and Land Cover Change (LUCC) datasets, exemplified by the History Database of the Global Environment (HYDE), the Sustainability and the Global Environment (SAGE), the Pongratz Julia (PJ) and the Kaplan and Krumhardt 2010 (KK10) (Goldewijk et al., 2017; Kaplan et al., 2011; Pongratz et al., 2008; Ramankutty et al., 2008; Ramankutty and Foley, 1999), have been extensively employed in global change research. Such as carbon emission and carbon neutrality (Xu et al., 2024), climate data construction (Gortan et al., 2024), ecological footprint (Wang et al., 2024), and biological population assessment (Ye et al., 2024), etc. Furthermore, with the progress of research, historical LUCC study outcomes pertaining to the Northeast China have proliferated from a global scale down to the county level (Bai et al., 2007; Cao et al., 2021; He et al., 2023; Hurtt et al., 2020; Jia et al., 2023; Li et al., 2016; Li et al., 2018; Wu et al., 2020; Wu et al., 2022; Yang et al., 2017; Ye et al., 2009; Ye and Fang, 2012; Yu et al., 2021; Zhang et al., 2014; Zhang et al., 2022; Zeng et al., 2011; Tian, 2005; Jin et al., 2015; Shi, 2015; Zhang, 1991; Zhou, 2001). However, a disparity or uncertainty persists in the standardization and spatiotemporal accuracy of the aforementioned cropland data, leading to conflicts arise between datasets and historical evidence of regional agricultural development. Consequently, enhancing the accuracy and credibility of historical LUCC datasets remains a focal point in international LUCC research (Gaillard et al., 2018; Yang et al., 2024; Yu et al., 2021). Reconstructing relatively accurate historical cropland cover at the basic-level administrative divisions based on actual historical agricultural development is a primary method for improving historical LUCC datasets (Goldewijk et al., 2017; Yu et al., 2021). For instance, the HYDE dataset demonstrates a boundary effect influenced by modern provincial administrative divisions in Northeast China, resulting in discontinuities in the spatial distribution of cropland in regions within the same historical agricultural development process. Considering the historical evolution of administrative divisions in China (Zhao et al., 2024), the cropland of smallest administrative divisions that can be reconstructed at present is the county-level, which suggests that it is possible to control the error of the gridded allocation to between $0.5° \times 0.5°$ and $1° \times 1°$. Therefore, long-term precise cropland area change datasets with basic-level administrative divisions and standardized time-points will not merely improve the accuracy and credibility of global historical LUCC datasets, but will also play a crucial role in enhancing the precision of climate and environmental simulations and supporting detailed environmental effect analyses in Northeast China.

Northeast China is one of the most important grain bases in China today. The grain output constitutes 25.18% of the national total, with corn and soybean contributing 41.64% and 56.20%, respectively (National Bureau Of Statistics, 2023). A

study has indicated that the supply centers for China's three major grains (wheat, corn, rice) significantly moved to the Northeast from 2000 to 2020, while the demand centers did not move simultaneously. This shift underscores the rapidly increasing importance of the Northeast China in ensuring China's food security (Xuan et al., 2023). Furthermore, the majority of China's black soil is distributed in Northeast China, which provides an important foundation for the productivity of crops. A study has pointed out that compared to other global black soil regions, the Northeast black soil region's yields of eight major crops (excluding rice) remained in the top three among the world's main black soil distribution countries from 2000 to 2015, with Russia and Ukraine occupying the first two positions (Wang et al., 2024). Additionally, long-term precise cropland area change data reflects the significance for soil and water conservation research in Northeast China, thereby ensuring food security. A typical case study in the Northeast China examined the long-term effects of cultivation on soil carbon, nitrogen, and bacterial community in the Northeast black soil region. The results indicated that prolonged cultivation (e.g., 152 years) led to a negatively and exponentially decline in SOC and total nitrogen (Liu et al., 2024).

The dataset in this study presents a critical update and extension of the former historical cropland cover change in the three provinces of Northeast China over the past 300 years (Ye et al., 2009). Throughout the prolonged agricultural development, the natural vegetation landscape in the Northeast China has undergone notable transformations. In this study, we used the improved historical cropland reconstruction methods to reconstruct 28 time-points cropland area by assimilating multiple data sources in Northeast China from 1000 to 2020. The mainly new features of this dataset include: (1) Extended the reconstruction period to 1000~2020, aligning with the standard time-points of internationally established global historical LUCC datasets; (2) the reconstruction included the entire East of Inner Mongolia, which area accounts for approximately 45% of the Northeast China. (3) the smallest administrative divisions for the reconstructed cropland are at the provincial-level from 1000 to 1600, and at the county-level from 1700 to 2020. Our main objective is to provide a long-term time series of cropland area change dataset in Northeast China that is close to the historical "truth value" under a unified standard.

**2 Data and methods**

**2.1 The study area and the framework for cropland reconstruction**

The definition of Northeast China in this study includes Heilongjiang, Jilin and Liaoning Provinces, Hulunbuir City, Hinggan League, Tongliao City, Chifeng City and Xilin Gol League of Inner Mongolia. Northeast China is located between 38°43' and 53°33' N and between 111°59' and 135°05' E, with a total area of approximately $1.45 \times 10^6$ km$^2$, about 15.1% of the total area of China, and the main part of Northeast China has a temperate continental monsoon climate. In this study, the seven time points from 1000 to 1600 are reconstructed based on the provincial-level administrative districts and derived from the Historical Atlas of China (Tan, 1982a; Tan, 1982b). For the period from 1700 to 2020, twenty-one time points are reconstructed based on the county-level administrative districts using the 1:1,000,000 public version of basic geographical information data

released by the National Geomatics Center of China (2021 edition)
(https://www.webmap.cn/commres.do?method=result100W, last access: 10 January 2024). For the sake of convenience in
research and considering the historical evolution of each region, this study consolidates the administrative districts under each
prefecture-level city in the Northeast China into a single administrative unit. Additionally, Nianzishan District is merged into
Longjiang County, Bayuquan District into Gaizhou City, Qingmenhe District into Fuxin County, Qinghe District into Kaiyuan
City, Zhanqian District into Dashiqiao City, Zhalainuoer District into Manzhouli City, Huolinguole City into Zhalute Banner,
and Aershan City into Horqin Right Wing Front Banner.

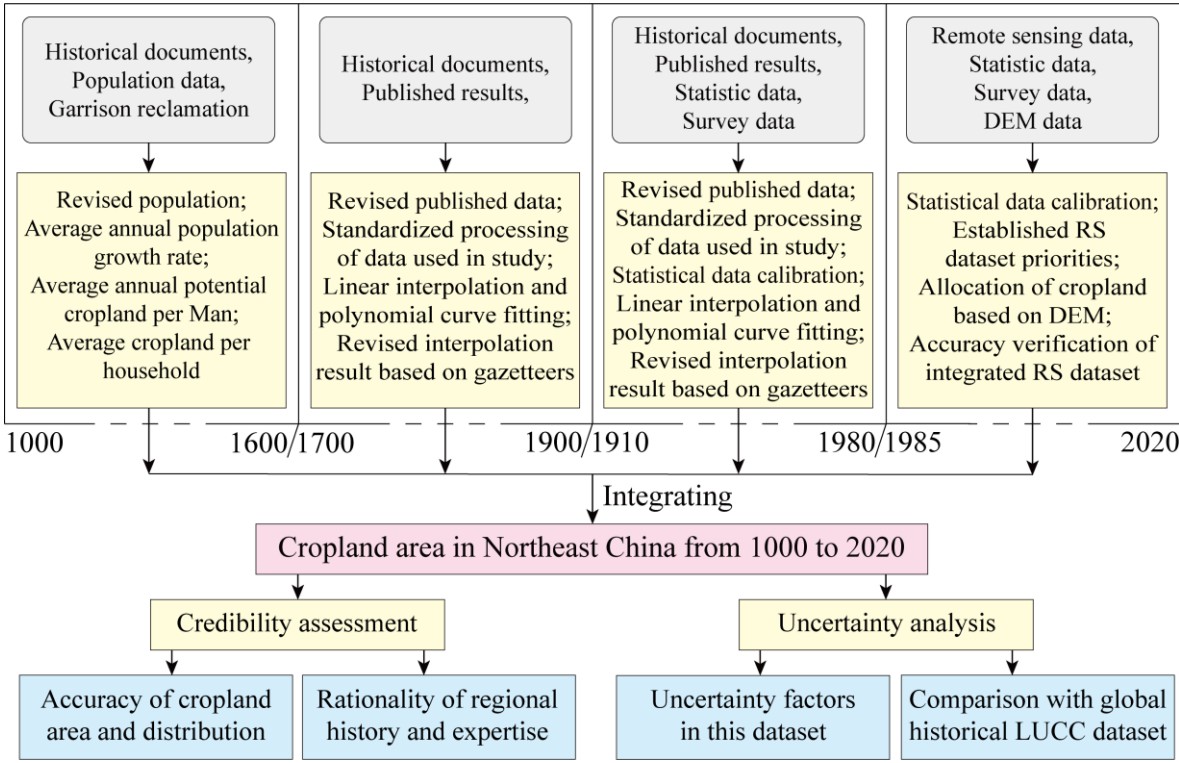


**Figure 1: The framework for reconstructing cropland area of Northeast China from 1000 to 2020.**

The framework of the cropland data reconstruction process in this study is illustrated in Fig. 1. It is essential to note that,
unlike reconstructing historical cropland through simulation or speculation, the data foundation in this study incorporates
historical literature, proxy data such as population data, revised published results, statistical data, survey data, and remote
sensing data products. Historical period reconstruction primarily relies on population data from historical time points.
Population data for adjacent standard time points are calculated using the average annual growth rate, and proxy indicators
such as average annual cropland area per Man and average cropland area per household are employed to calculate cropland
area. Additionally, after correcting published data and supplementing blank areas through standardized data processing, we
used historical facts to interpolate cropland area from nearby time points to standard time points through linear interpolation.
Trend extrapolation and total control are achieved through polynomial curve fitting. Finally, errors that may exist in the
interpolation are corrected based on local gazetteers of China (https://fz.wanfangdata.com.cn/, last access: 10 January 2024).
The reconstruction in the modern period primarily involves analyzing the linear relationship between statistical data and survey
data. Survey data sequences established are used to control the cropland pixel data obtained through the regional-scale
constrained integration of remote sensing data.
**2.2 Data sources and reconstruction methods**
**2.2.1 Reconstruction of cropland area from 1000 to 1600**
This study covers seven standard time points from 1000 to 1600, spanning the Liao, Jin, Yuan, and Ming dynasties. Due to the
absence of direct records of cropland area during this period, cropland reconstruction primarily relies on historical documents,
population data, and garrison reclamation data corresponding to the provincial-level administrative districts. During the Liao
Dynasty period, this study based on the Dynastic History of Liao Dynasty and the History of Population in China (Wu and Ge,
2005a; Toqto'A, 1974) along with other published results (Ge, 2002; Han, 1999; Tan, 1982b), to reconstruct the agricultural
and non-agricultural populations within five provincial-level administrative districts in 1111, with an average household size
of 6.5 people, 2.08 of whom were Man (a male between the ages of 15 and 50 years in the Liao Dynasty). Population data for
the five districts in 1000 and 1100 were calculated based on a 0.5% average annual population growth rate (Wu and Ge, 2005a).
During the Jin Dynasty period, this study is primarily based on the Dynastic History of Jin Dynasty and the History of
Population in China (Wu and Ge, 2005a; Toqto'A, 1975) along with other published results (Li et al., 2018; Han, 1999; Jin and
Mikami, 1984; Liu, 1994a; Liu, 1994b; Tan, 1982b), to reconstruct the agricultural and non-agricultural populations within
five provincial-level administrative districts in 1207, with an average agricultural household size of 5.96 people, 2 of whom
were Man (a male between the ages of 17 and 59 years in the Jin Dynasty), while an average non-agricultural household size
of 10.59 people. Population data for the five districts in 1200 were calculated based on a 0.9% average annual population
growth rate (Toqto'A, 1975).
When calculating cropland area during the Liao and Jin period (1000~1200), this study primarily involves adjusting the
agricultural and non-agricultural population quantities to standard time points. Combining with the constructed method of the
average annual cropland area per Man for agricultural population and the average cropland area per household for non-
agricultural population during the Liao and Jin Dynasties (Jia et al., 2023), the cropland areas for provincial-level
administrative units in the Northeast China in the 1000, 1100, and 1200 are calculated separately (Table 1). The main algorithm
applied in the Liao and Jin Dynasties can be found in the supplementary materials. Furthermore, due to the lack of significant
technological changes in agricultural production in the Northeast China and the southward shift of the northern boundary of
the farming-pastoral ecotone within the study area (He et al., 2023; Han, 2012; Zhang et al., 1997), this study maintains
consistency with the Liao and Jin Dynasties. The average annual cropland area per Man for agricultural population is set at 14
*Mu* (0.93 hm$^2$), and the average cropland area per household for non-agricultural population is set at 2 *Mu* (0.13 hm$^2$) during

142 the Yuan and Ming Dynasties (1300~1600).

143  During the Yuan Dynasty, this study primarily based on the Dynastic History of Yuan Dynasty (Song, 1976) to obtain the

144 garrison reclamation area and corresponding number of soldiers in the Northeast China around 1300, and the average cropland

145 area per garrison soldier is 100.1 *Mu* (6.67 hm$^2$). Additionally, based on the Dynastic History of Yuan Dynasty and the History

146 of Population in China (Wu and Ge, 2005a; Cao and Ge, 2005b; Song, 1976) along with other published results (Cong, 1993a;

147 Zhan, 2017; Xue, 2012; Zhou, 2021), this study reconstructs the number of ordinary households and Mongol households within

148 the three provincial-level administrative districts of the study area during the Yuan Dynasty (Tan, 1982a). Ordinary households

149 are further divided into Han households (agricultural population) and other minority ethnic households (non-agricultural

150 population) in a 7:3 ratio (Cong, 1993b), with an average agricultural household size of 5 people, 1.67 of whom were Man (a

151 male between the ages of 15 and 59 years in the Yuan Dynasty). Population data for garrison soldiers, Han households, minority

152 ethnic households, and Mongol households in the three districts around 1300 are calculated based on different average annual

153 population growth rates ranging from 0.6% to 1.8% during the Yuan Dynasty (Wu and Ge, 2005a). After obtaining the

154 population data, this study subtracts the garrison soldiers and their corresponding households from the ordinary households.

155 Subsequently, the remaining ordinary households are divided into Han households and minority ethnic households in a 7:3

156 ratio. The cropland area for agricultural population is calculated based on the average annual cropland area per Man for

157 agricultural population, while the cropland area for non-agricultural population, including Mongol households, is calculated

158 using the average cropland area per household for non-agricultural population referring the Liao and Jin Dynasties (Table 1).

159  During the Ming Dynasty, this study primarily based on the Dynastic History of Ming Dynasty (Zhang, 1974) to obtain

160 the garrison reclamation area in the Northeast China around 1400. According to historical records and verification, it is

161 determined that each garrison soldier in the Liaodong region possessed 46 *Mu* (3.07 hm$^2$) of cropland, with the proportion of

162 garrison soldiers among soldiers being approximately 30%, and the number of dependents for each soldier being twice that of

163 soldiers (Cao and Ge, 2005b; Li, 2019; Wang, 2009; Zhang, 1974). Additionally, based on the Dynastic History of Ming

164 Dynasty and the History of Population in China (Cao and Ge, 2005b; Zhang, 1974) along with other published results (Cong,

165 1985; Kong and Feng, 1989; Li, 2019; Tan, 1982a), this study reconstructs the population of soldiers and their dependents,

166 ordinary households/aborigines, and the population of minority ethnic households and Mongols (non-agricultural population)

167 within the four provincial-level administrative districts in the 1400. Referring to historical records such as refugee migration,

168 the construction of the Great Wall, and supplementary border garrisons (Cao and Ge, 2005b; Kong and Feng, 1989; Liu et al.,

169 2016; Tan, 1982a), the historical maps for the 1500 and 1600 are divided into three provincial districts, and the population data

170 for these two time points is obtained based on the aforementioned historical documents. During this period, all regular soldiers

171 in the Dusi of Eastern Liao and one-third of their dependents would operate cropland as farmers. The average agricultural

172 household (ordinary households/aborigines/refugees/migrants) size of 6, 2.25 of whom were Man (a male between the ages of

16 and 60 years in the Ming Dynasty) in the Dusi of Eastern Liao. The average non-agricultural household (minority ethnic households) size of 6, 2 of whom were Man in the Dusi of Nuergan, while size of the Mongol households is 5, 1.67 of whom were Man. Population data for soldiers and their dependents, ordinary households/aborigines/refugees/migrants, minority ethnic households in the Dusi of Nuergan, and Mongol households in the western part of the study area in the three provinces are calculated for the 1500 and 1600 based on average annual population growth rates of 0.8%, 0.5%, 0.2%, and 0.15%, respectively (Cao and Ge, 2005b). After obtaining the population data, we calculated the garrison reclamation area and civilian cropland area within the Dusi of Eastern Liao and the Dusi of Beiping based on the population of soldiers and agricultural population (ordinary households/aborigines) in the 1400. The minority ethnic population in the Dusi of Nuergan and the Mongol population in the Dada are calculated as non-agricultural population referring the Liao and Jin Dynasties (Table 1). For the 1500 and 1600, we calculated the garrison reclamation area and civilian cropland area within the Dusi of Eastern Liao based on the population of soldiers and agricultural population (ordinary households/aborigines/refugees/migrants). The minority ethnic population in the Dusi of Nuergan and the Mongol population in the Dada are calculated as non-agricultural population referring the Liao and Jin Dynasties (Table 1). The main algorithm applied in the Yuan and Ming Dynasties can be found in the supplementary materials.

**Table 1: The index of cropland area reconstruction from 1000 to 1600**

| Period | Population type | Population ($10^4$) | Proportion of household registration | Corresponding cropland area | Total cropland area (km$^2$) |
|---|---|---|---|---|---|
| 1000, 1100 | Agricultural population | 371(1000); 612(1100) | Average household size: 6.5 people, 2.08 of whom were Man | Average annual cropland area per Man is 14 *Mu* (0.93 hm$^2$) | 5513(1000); 9078(1100) |
| | Non-agricultural population | 140(1000); 231(1100) | | Average cropland area per household is 2 *Mu* (0.13 hm$^2$) | |
| 1200 | Agricultural population | 587 | Average household size: 5.96 people, 2 of whom were Man | Average annual cropland area per Man is 14 *Mu* (0.93 hm$^2$) | 16949 |
| | Non-agricultural population | 338 | Average household size: 10.59 people | Average cropland area per household is 45.3 *Mu* (3.02 hm$^2$) | |
| 1300 | Garrison soldiers | 0.8 | Each soldier represents a household | Average per garrison soldier is 100.1 Mu (6.67 hm$^2$) | 4350 |
| | Agricultural population | 111 | Average household size: 5 people, 1.67 of whom were Man | Average annual cropland area per Man is 14 *Mu* (0.93 hm$^2$) | |
| | Non-agricultural population (Minority ethnic household) | 137 | | Average cropland area per household is 2 *Mu* (0.13 hm$^2$) | |
| 1400 | Soldiers and their dependents | 70 | Approximately 30% of garrison soldiers; Soldiers : dependents = 1 : 2 | Average per garrison soldier is 46 *Mu* (3.07 hm$^2$) | 2790 |
| | Agricultural population (ordinary | 10 | Average household size: 6 people, 2.25 of whom were | Average annual cropland area per Man is 14 *Mu* (0.93 hm$^2$) | |

| | | | Man | | |
|---|---|---|---|---|---|
| | Non-agricultural population (Minority ethnic household, Mongol household) | 40 | Average minority ethnic household size: 6 people, 2 of whom were Man; Mongol household size: 5, 1.67 of whom were Man | Average cropland area per household is 2 $Mu$ (0.13 hm$^2$) | |
| 1500, 1600 | Soldiers and their dependents | 25(1500); 12(1600) | Approximately 30% of garrison soldiers; Soldiers : Dependents = 1 : 2 | Average per garrison soldier is 46 $Mu$ (3.07 hm$^2$); Regular soldiers and one-third of their dependents is 14 $Mu$ (0.93 hm$^2$) | 4875(1500); 5868(1600) |
| | Agricultural population (ordinary households/aborigines/ refugees/migrants) | 83(1500); 137(1600) | Average household size: 6 people, 2.25 of whom were Man | Average annual cropland area per Man is 14 $Mu$ (0.93 hm$^2$) | |
| | Non-agricultural population (Minority ethnic household, Mongol household) | 68(1500); 81(1600) | Same as 1400 | Average cropland area per household is 2 $Mu$ (0.13 hm$^2$) | |

189

### 2.2.2 Reconstruction of cropland area from 1700 to 1900

The reconstruction of cropland in this study at five standard time-points from 1700 to 1900 is primarily based on published results and historical documents. Among them, published results utilize the county-level cropland fraction data (CNEC) reconstructed by Ye (Ye et al., 2009) for the three provinces in Northeast China in 1683, 1735, 1780, and 1908. Additionally, data on cropland fraction for 15 counties and districts, including Chifeng City, Balinzuo Banner, Balinyou Banner, Linxi County, Wengniute Banner, Kalaqin Banner, Ningcheng County, Aohan Banner, Kulun Banner, Naiman Banner, Taipusi Banner, Xianghuang Banner, Zhengxiangbai Banner, Zhenglan Banner, and Duolun County, reconstructed by Tian (Tian, 2005), are available for the years 1724, 1782, 1868, and 1911. Detailed description of the data and methods for these published results can be found in the supplementary materials.

Before utilizing the published results, this study examined and corrected issues present in the data, unifying it onto the base map used in this study. (1) Correction of published results: CNEC data (Ye et al., 2009) was adjusted based on the historical evolution of administrative boundaries to modern county-level administrative units. In 1908, cropland areas were missing for Qian Gorlos Mongolian Autonomous County, Jiaohe City, Yanji City, Wangqing County, Huichun City, Helong City, and Huinan County in Jilin Province. Wu (Wu, 2021) interpolated these missing values using the principles of geographical proximity and similarity in the regional agricultural development stage. By following the above method, we interpolated data for problematic counties in Jilin Province from CNEC data using settlement names evolution data for the past 300 years (Zeng et al., 2011). It is worth noting that for certain time points, due to the absence of cropland in neighboring counties, this study adopted the approach of multiplying the cropland area owned by unit settlements within Jilin Province at

that time by the number of settlements in the respective county to obtain the cropland area (Table S1). Furthermore,
discrepancies were identified in used CNEC data for some counties in Heilongjiang and Liaoning provinces compared to
published data. This study corrected these inconsistencies after verifying historical documents (Table S1).
(2) Unified administration boundaries: The CNEC data (Ye et al., 2009) in 1683, 1735, and 1780 corresponds to historical
Qing Dynasty administrative districts, and the administrative districts used in 1908, 1914, 1931, 1940, 1950, and 1980 also
differed from that of this study. The approach taken in this study involves unifying the cropland fraction within each county or
district. The modern county-level administrative vector map used in this study is overlaid with Ye's county-level cropland
fraction map. Then we calculated the area of overlap between each county or district in this study and Ye's corresponding
county or district and then calculates the cropland area based on the proportional statistics. Similarly, for the Tian's data (Tian,
2005) used in this study for cropland fraction in 1724, 1782, 1868, 1911, and 1933, the same method is applied to unify them
onto the modern map used in this study.
(3) Linear interpolation and polynomial curve fitting to obtain the cropland area: Previous studies have used the linear
interpolation and polynomial curve fitting to reconstruct cropland areas (He et al., 2017; Jin et al., 2015; Ramankutty and Foley,
1999; Wei et al., 2016; Wei et al., 2021; Ye et al., 2015; Yu, 2019; Fang et al., 2021), and the interpolated data did not reduce
the credibility of their datasets. In addition, previous studies have shown that in the process of reclamation in the Northeast
China over the past 300 years, 1860 was a dividing point between slow growth and rapid growth, mainly due to the
implementation of the immigration and reclamation policy by the Qing government (Fang et al., 2020; Ye et al., 2009; Fang et
al., 2005; Kong and Feng, 1989). Therefore, this study selected the CNEC data (Ye et al., 2009) in 1683, 1735, 1780, 1908 and
1914 for linear interpolation and polynomial curve fitting of cropland area data for each county or district in the three provinces
of Northeast China, obtaining data for 1700, 1750, 1800, 1850 and 1900. In addition, this study selected the data from Tian
(Tian, 2005) in 1724, 1782, 1868, and 1911; the CNEC data (Ye et al., 2009) in 1735; the data from Ye (Ye and Fang, 2012)
in 1916 for linear interpolation and polynomial curve fitting to obtain cropland area data for 1700, 1750, 1800, 1850, and 1900
in the Eastern of Inner Mongolia. The problems that may be encountered during the operation and the corresponding solutions
are as follows:
①Linear interpolation and determination of zero values. The time points involved in this issue include 1700 and 1750 for
the three provinces of Northeast China; 1750, 1800, and 1850 for East of Inner Mongolia. For instance, in Northeast China,
the cropland area in each county in 1700 is interpolated based on records from 1683 and 1735. At 1700, there are no negative
values, but there may be zero values. Specifically, the cropland value in 1683 is 0, while there is definite value in 1735. Our
solution involves consulted contemporary county gazetteers to verify the history of land reclamation in 1700. If so, a
polynomial curve fitting trend extrapolation was applied to obtain the proportional relationship at the provincial level for
adjacent points on the extrapolated trend. Then this proportion was multiplied by the cropland area of the county at the adjacent
time-point to obtain the cropland area at that time-point. If the land was not reclaimed, the value at that time point was
considered as zero. Similarly, other counties involved in interpolation adopt the same solution when encountering this situation.
②Polynomial curve fitting and correction of negative values. Besides the previously mentioned linear interpolation,
polynomial curve fitting based on the least squares method may encounter problems with data points resulting in negative
values. First of all, the main reason for this issue is our historical determination that 1860 was a dividing point between slow
and rapid growth. Therefore, we use 1860 as a breakpoint and separate interpolated the data for Ye (Ye et al., 2009, Ye and
Fang, 2012) and Tian (Tian, 2005) before and after this period. Second, for time points that cannot be directly obtained through
linear interpolation, cropland need to be calculated by polynomial fitting backwards (1800 and 1850 in the three provinces of
Northeast China; 1900 in East of Inner Mongolia). For instance, in Northeast China, cropland area in each county in 1800 and
1850 are derived from data in 1683, 1735, and 1780 using polynomial curve fitting method. Some counties may show a decline
in cropland, potentially resulting in negative values in the extrapolation results. Our solution involves using the proportion of
provincial administrative level to multiply by the cropland area in 1780 for correction in the counties' cropland area in 1800
and 1850. Third, for time points that cannot be directly obtained through linear interpolation, cropland need to be calculated
by polynomial fitting forwards (1900 in the three provinces of Northeast China; 1700, 1910 in East of Inner Mongolia). For
instance, in Northeast China, cropland area in each county in 1900 is derived from data in 1908 and 1914 using polynomial
curve fitting method. Due to rapid growth of cropland in some counties from 1908 to 1914, the extrapolation for 1900 may
result in negative values. Our solution involves using the proportion of provincial administrative level to multiply by the
cropland area in 1908 for correction in the counties' cropland area in 1900.
It should be noted that, considering the historical development process of Northeast China during the Qing Dynasty, war
factors, and the encouraging land reclamation policies implemented by the Qing government after 1860, we determined that
the cropland area in each county of Northeast China in 1900 would not significantly exceed that of 1908. During this period,
in Northeast China, the total cropland area was gradually increasing and was not significantly affected by events such as the
Second World War, which led to a notable decrease in cropland area in 1950 compared to 1930 and 1940. Therefore, when the
extrapolated value for a county in 1900 exceeds that of 1908, the proportion of provincial administrative level is used to
multiply by the cropland area in 1908 for correction in the county's cropland area in 1900.
③The determination of initial cultivation occurred between 1780 and 1908. Few counties in Northeast China where
cropland was zero in 1683, 1735, and 1780, but had cropland in 1908. Our solution involves consulted contemporary county
gazetteers to verify the history of land reclamation between 1800 and 1900. If local gazetteers indicate the initial cultivation
occurred before 1860, this study applies the same method as described in "①Linear interpolation and determination of zero
values". If the initial cultivation began after 1860, this study applies the same method as described in "②Polynomial curve
fitting and correction of negative values". All the counties where this situation occurs can be found in Table S2.

### 2.2.3 Reconstruction of cropland area from 1910 to 1980

The reconstruction of cropland at eight standard time points from 1910 to 1980 in this study is mainly based on published results, historical documents, statistical data, and survey data. Among these, the published results include the cropland fraction for the three provinces in Northeast China in 1908, 1914, 1931, 1940, 1950, and 1980 (CNEC) (Ye et al., 2009). As well as the cropland fraction for the farming-pastoral ecotone area reconstructed by Ye in 1916 and 1940 (Ye and Fang, 2012). Additionally, Tian's reconstruction provides cropland fraction for 15 counties in the Eastern of Inner Mongolia in 1911 and 1933 (Tian, 2005). Historical documents include the Summary of county governance in Northeast China (Xiong, 1933) to supplement cropland area data for the Eastern of Inner Mongolia in 1931. Statistical data include Agricultural and Animal Husbandry Production Statistics (Inner Mongolia Provincial Bureau Of Statistics, 1983) to obtain county-level cropland area for the Eastern of Inner Mongolia in 1950, 1960, 1970, and 1980. Survey data include Manchuria Economic Statistics Charts (Office Of The Governor-General Of Kwantung, 1918) to obtain prefecture-level cropland area data for the Eastern of Inner Mongolia in 1917 as a reference. The North Manchuria and East Support Railway (East Branch Railway Administration Of Russia and South Manchuria Railways Co., 1923) is used as survey data to supplemented for various counties in the Eastern of Inner Mongolia in 1911 and 1914, which was not covered by existing data from Ye and Tian. Additionally, a digital version of the Manchuria Political Map from this document was used to obtain county-level district maps for Northeast China in the 1920s. Detailed description of the data and methods for these published results can be found in the supplementary materials.

Before using the published data from this period, this study also assessed and corrected the issues present in the data. Additionally, when supplementing the data using historical documents, statistical data and survey data, this study referred to the data processing methods of the aforementioned published studies. (1) Correction of published results: This study has provided specific explanations for the correction of CNEC data for this period in previous sections, as detailed in Table S1.

(2) Standardization of Data: This study adopted the processing method used by Ye (Ye et al., 2006) for the Summary of county governance in Northeast China (Xiong, 1933). It converted the Qing Dynasty's *Mu* unit to the standard unit of measurement, square kilometers ($km^2$), and made a 10% correction to align this data with the survey data. For the Manchuria Economic Statistics Charts and the North Manchuria and East Support Railway (Office Of The Governor-General Of Kwantung, 1918), this study followed Ye's (Ye et al., 2006) analysis method for similar survey data, treating it as the actual cropland area. Regarding the standardization of administrative boundaries, this study utilized the digitized Manchurian Political Map and employed the method aforementioned to map it onto the modern administrative boundary map used in this study. The standardization of measurement units followed the conversion from the measurement units used in the Japanese survey data to the universal unit of measurement, square kilometers ($km^2$), as per Weights and Measures in Northeast China (South Manchuria Railways Co., 1927).

(3) Correlation analysis between statistical data and survey data: In this study, we referred the method used by Ye (Ye et

al., 2009) in analyzing statistic data for the simultaneous period in the three provinces in Northeast China to process the county-level cropland area statistical data for the 1950, 1960, 1970, and 1980 in the Eastern of Inner Mongolia (Inner Mongolia Provincial Bureau Of Statistics, 1983). It is found a stronger correlation between the statistical data and land survey data in 1985 (National Bureau Of Statistics, 1989; Committee Of Integrative Survey Of Natural Resources and Committee Of National Planning Of Chinese Academy Of Sciences, 1990), with a linear regression equation of y=1.3234x and $R^2$=0.9851 (Fig. 2). That means the land survey data in the Eastern of Inner Mongolia is approximately 32.34% higher than the corresponding statistical data, then corrected cropland area data by 32.34% for each county in the Eastern of Inner Mongolia for the 1950, 1960, 1970, and 1980.

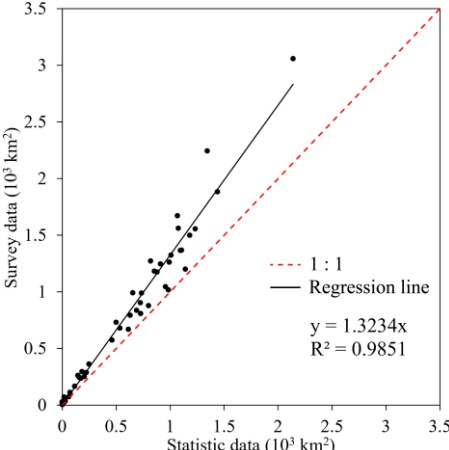

**Figure 2: Correlation between the statistical cropland data and survey cropland data of the counties in the Eastern of Inner Mongolia in 1980's.**

(4) Linear interpolation and polynomial curve fitting to obtain the cropland area: This study selected CNEC (Ye et al., 2009) data in 1908 and 1914 for linear interpolation and polynomial curve fitting of cropland area data for each county or district in the three provinces of Northeast China, obtaining data for 1910 and 1920. Additionally, this study selected the data from Tian (Tian, 2005) in 1911 and the data from Ye (Ye and Fang, 2012) in 1916 and 1940, and the corrected data in 1931 from Summary of county governance in Northeast China (Xiong, 1933) for linear interpolation and polynomial curve fitting of cropland area data for each county or district in the Eastern of Inner Mongolia, obtaining data for 1910 and 1920. Since the following operations are the same as 1700~1900, and the problems that may be encountered during the operation and the corresponding solutions have been detailed above, it will not be repeated here.

It should be noted that this study considers the corrected data in 1931 in various counties of the Northeast China as data for 1930. In addition, the cropland area data for the year 1940 mainly based on the corrected published results. For the missing data in single-digit counties of the Eastern of Inner Mongolia, this study uses data recorded in local gazetteers to fill in the gaps.

**2.2.4 Reconstruction of cropland area from 1985 to 2020**

The reconstruction of cropland in this study from 1985 to 2020 at eight standard time points is primarily based on remote sensing data products, statistical data, survey data, and DEM data. Among these, eight sets of remote sensing data products were used (Table 2): AGLC (Xu et al., 2021), CLDC (Yang and Huang, 2021), ESA_WorldCover (Zanaga, 2021), Esri_LandCover (Karra et al., 2021), FROM_GLC (Gong et al., 2013), GFSAD30 (Thenkabail et al., 2021), GLC_FCS30 (Zhang et al., 2023), GlobeLand30 (Chen et al., 2015). It is worth mentioning that we conducted research on ESA_WorldCover and Esri_LandCover after resampling them to a resolution of 30 meters. Survey data includes the year 1985 county-level first general land investigation (Committee Of Integrative Survey Of Natural Resources and Committee Of National Planning Of Chinese Academy Of Sciences, 1989), provincial-level data from the first national land survey (Li, 2000), prefecture-level data from the second national land survey, and county-level data from the third national land survey (https://gtdc.mnr.gov.cn/Share#/, last access: 10 January 2024).

**Table 2: Characteristics of the eight RS products**

| Product | Satellite Sensor | Type | Resolution | Year | Cropland Classes | URL | Reference |
|---|---|---|---|---|---|---|---|
| AGLC | Landsat 5 TM Landsat 7 ETM+ Landsat 8 OLI | Boolean | 30m | 2000, 2005, 2010, 2015 | 10.Cropland | https://code.earthengine.google.com/?asset=users/xxc/GLC_2000_2015 [2024/01/10] | (Xu et al., 2021) |
| CLDC | Landsat 8 OLI TM ETM+ | Boolean | 30m | 1985, 1990, 1995, 2000, 2005, 2010, 2015, 2019 | 1.Cropland | https://doi.org/10.5281/zenodo.4417810 [2024/01/10] | (Yang and Huang, 2021) |
| ESA_World Cover | Sentinel-1 Sentinel-2 | Boolean | 10m | 2020 | 40.Cropland | https://viewer.esa-worldcover.org/worldcover/ [2024/01/10] | (Zanaga, 2021) |
| Esri_LandCover | Sentinel-2 | Boolean | 10m | 2020 | 5.Crops | https://livingatlas.arcgis.com/landcover/ [2024/01/10] | (Karra et al., 2021) |
| FROM_GLC | Landsat TM, ETM+, OLI | Boolean | 30m | 2010, 2015 | 10.Cropland | https://data-starcloud.pcl.ac.cn/zh [2024/01/10] | (Gong et al., 2013) |
| GFSAD30 | Landsat ETM+ OLI | Boolean | 30m | 2015 | 2.Cropland | https://lpdaac.usgs.gov/products/gfsad30aunzcnmocev001/ [2024/01/10] | (Thenkabail, 2021) |
| GLC_FCS30D | Landsat OLI | Boolean | 30m | 1985, 1990, | 10.Rainfed cropland | https://zenodo.org/records/8239305 [2024/01/10] | (Zhang et al., 2023) |

| | | | | 1995, 2000, 2005, 2010, 2015, 2020 | 11.Herbaceous cover 12.Tree or shrub cover (Orchard) 20.Irrigated cropland | | |
|---|---|---|---|---|---|---|---|
| GlobeLand30 | Landsat TM/ETM+, HJ-1 | Boolean | 30m | 2000, 2010, 2020 | 10.Cropland | http://www.webmap.cn/map DataAction.do?method=glo balLandCover [2024/01/10] | (Chen et al., 2015) |

In this study, based on remote sensing data products, statistical data, survey data, and DEM data, we have developed a constrained integration method that combines multisource cropland cover products with survey data. (1) Correlation analysis between statistical data and survey data: This study obtained cropland survey data at the county-level in 1985, at the provincial-level in 1996, at the prefecture-level in 2010 and 2015, and at the county-level in 2020. It should be noted that the 2015 cropland survey data was obtained through the annual land change survey based on the second NLS, which is relatively less accurate than the cropland areas from the standard time-points data after the nationwide surveys (e.g., 2010 and 2020). According to the Ministry of Natural Resources of the PR China, the annual land change survey is based on the results of the nationwide survey and the previous year's land change survey, examining the current status and changes in the land use class, location, area, and distribution of various urban and rural lands across the country at the end of each year. For the missing years 1990, 2000, and 2005, this study referred to the correlation analysis between modern survey data and statistical data (Ye et al., 2009; Cropland Research Group, 1992). This study selected survey data and statistical data from 2010, 2015, and 2020 within the study area, respectively, and established linear regression equation between them. The results showed that the linear regression equation was y=1.2824x in 2010, and $R^2$=0.9689; y=1.1572x in 2015, and $R^2$=0.9743; y=1.1455x in 2020, and $R^2$=0.9714 (Fig. 3, Table S4). This indicates a high correlation between the two types of data at the three time points, and the survey data is approximately 14.6% to 28.2% higher than the statistical data at the same period, with an average of about 20%, then corrected cropland area data by 20% for each county in the study area for the 1990, 2000 and 2005.

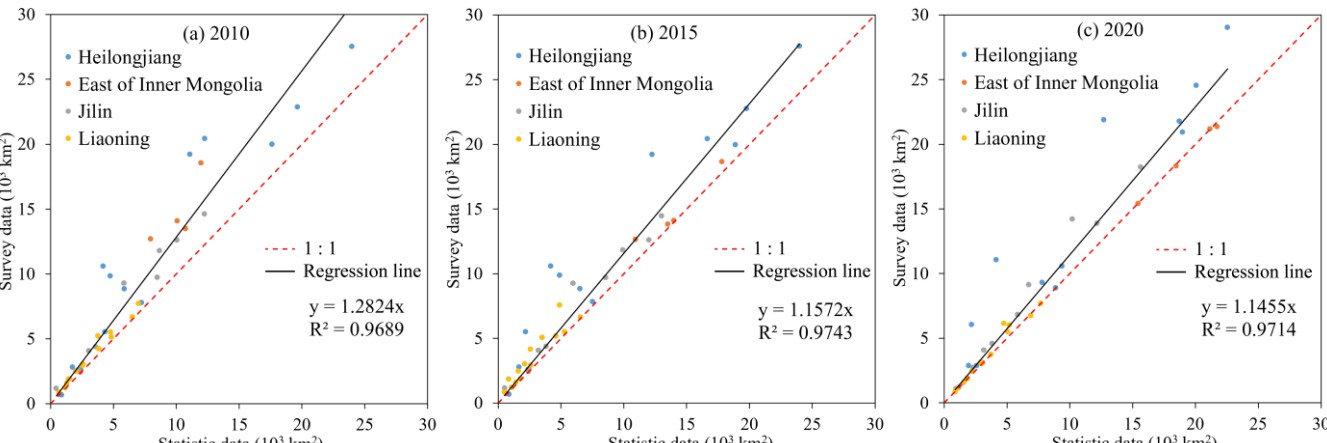

**Figure 3: Correlation between the statistical cropland data and survey cropland data at the prefecture-level in the Northeast China**

**in 2010, 2015 and 2020.**

(2) Establishing Dataset Priorities: After obtaining the modern land survey data levels for each province in the study area
at five-year intervals from 1985 to 2020, the difference between the cropland area in dataset i and the survey data on cropland
area, denoted as $D_{i,j}$, was calculated to evaluate the accuracy of the dataset, as shown in Equation (1):
$$D_{i,j}=abs\left(\frac{A_{s,j}-a_{i,j}}{A_{s,j}}\right) , \qquad (1)$$
where $A_{s,j}$ represents the survey data on cropland area in Northeast China for year j, and $a_{i,j}$ represents the cropland area in the
i-th subset of the land cover product for year j. The value of $D_{i,j}$ is lower when the consistency with survey data is higher,
indicating a higher priority for the input dataset. It should be noted that in this study, based on the priority and overlap of
remote sensing data products used at different time points, pixels in the study area are ranked. Pixels belonging to high-priority
products with high overlap will be prioritized as cropland.
(3) Allocation of cropland pixels based on DEM data: The survey data includes detailed slope classification, and the
slopes were categorized into five classes: <2°, 2~6°, 6~15°, 15~25°, and >25°, and the corresponding cropland areas for each
slope class were recorded. In this study, we selected NASA and METI's DEM data jointly released in 2019: ASTER Global
Digital Elevation Model V003 30m. The ASTER Global Digital Elevation Model V003 can be downloaded from the NASA
EARTHDATA website (https://www.earthdata.nasa.gov/, last access: 10 January 2024). Pixels prioritized as cropland were
allocated to the cropland area corresponding to each slope level in the survey data. The distribution results were controlled by
provincial-level cropland area survey data at different time points, resulting in the integration of cropland data at 30m resolution
for the Northeast China at 8 time points from 1985 to 2020.
(4) Accuracy assessment and validation of RS products integration results: This study utilizes the confusion matrix was
used to assess the accuracy of cropland products. The Producer Accuracy (P.A.) and User Accuracy (U.A.) for each product in
2020 are calculated as two indicators to evaluate the reliability of the spatial distribution of the cropland dataset. The calculation
methods are as follows:
$$P.A = \frac{X}{Ni} \times 100\% , \qquad (2)$$
$$U.A = \frac{X}{Nj} \times 100\% , \qquad (3)$$
where $X$ represents the number of correctly classified samples, $Ni$ represents the total number of verification samples, and $Nj$
represents the total number of samples in the classified result.
This study used three types of verification points for the verification of the integration result in year 2020 (Fig. S1): (1)
cropland sample points located in the study area from FROM-GLC. (2) 1052 sample points obtained through field
investigations conducted by the authors in April 2023 within the study area. (3) A total of 1200 random sample points were
generated within the study area. Using high-resolution imagery from Google Earth captured in 2020, the sample points were
visually interpreted and validated indoors through image comparison. The results show that the producer accuracy for cropland
pixels is 94.85%, and the user accuracy is 96.49% in year 2020. For non-cropland pixels, the producer accuracy is 91.12%,
and the user accuracy is 87.32%. The overall accuracy is relatively high.

**3 Results**

The cropland in Northeast China exhibited phase changes of expansion-reduction-expansion over the past millennium. The
cropland area in Northeast China increased from $0.55 \times 10^4$ km$^2$ in 1000 to $37.90 \times 10^4$ km$^2$ in 2020 and the average cropland
fraction increased from 0.37% to 26.27% (Fig. 4). Our results clearly show on the map the process of agricultural reclamation
in Northeast China and the expansion of cropland in the Songnen and Sanjiang Plains (Fig. 5).

**3.1 Changes in the historical cropland area in Northeast China over the past millennium**

The changes in cropland area in the Northeast China over the past millennium are illustrated in Figure 4. Overall, the proportion
of cropland area in the study area from 1000 to 1600 ranged from 0.74% to 4.5% of the total in 2020. During this period, from
1000 to 1200, the cropland area showed a growing trend, with an average annual growth rate of 0.56%. In 1200, it peaked at
$1.69 \times 10^4$ km$^2$, with an overall cropland fraction of 1.17%, although the cropland fraction across the region was relatively low.
From 1300 to 1600, the cropland area significantly decreased. In 1400, it reached the lowest point in the past millennium, at
$0.28 \times 10^4$ km$^2$, with an overall cropland fraction of only 0.19%. The average annual growth rate from 1400 to 1600 was 0.37%.
From 1600 to 1850, the cropland area grew slowly, with an average annual growth rate of 0.81%. During this period, the
proportion of cropland area in the study area increased from 1.55% to 11.52% of the total in 2020. After 1850, the cropland
area exhibited almost exponential growth. The agricultural area continued to expand northward, and this growth trend persisted
until 2020, with an average annual growth rate of 1.28%.

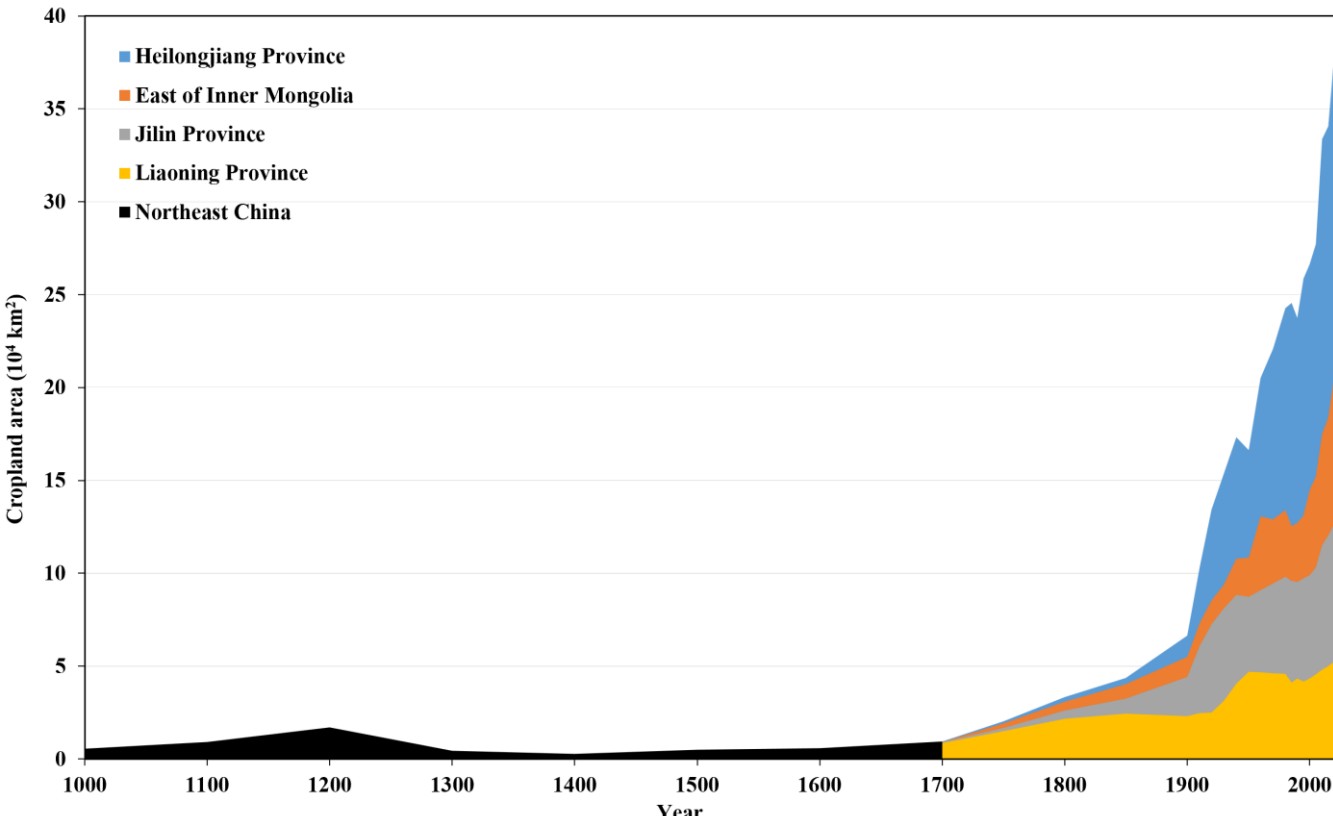

**Figure 4: Changes in total cropland area in the Northeast China from 1000 to 2020.**

**3.2 Spatial patterns of cropland distribution in Northeast China over the past millennium**

The changes in pattern of cropland in the Northeast China over the past millennium are shown in Figure 5. From 1000 to 1200, cropland in the study area had already reached a certain scale in spatial extent, mainly distributed in the Songliao Plain, especially in the southern part of the Liaohe Plain. The extent of cropland was roughly equivalent to the modern era. From 1300 to 1600, the main cultivation areas of cropland gradually receded southward to within the boundaries of Liaoning Province. From 1700 to 1850, cropland was mainly concentrated in the Liaoning Province. With the Qing government establishing military garrisons in the northern part of the Northeast China, farming areas was formed around these garrisons, and the farming area showed a trend of expanding northward. Due to the Qing government abandoning reclamation restrictive policies, from 1900 to 1950, the farming area gradually expanded to cover the entire region. Meanwhile, the cultivation intensity in the Hulunbuir City and Xilin Gol League of Inner Mongolia remained relatively low, influenced by war, leading to a slight decrease in the overall cropland fraction in 1950. After 1950, the farming area expanded rapidly and gradually formed a high cropland fraction agricultural zone with the Liaohe Plain, Songnen Plain, and Sanjiang Plain as its core.

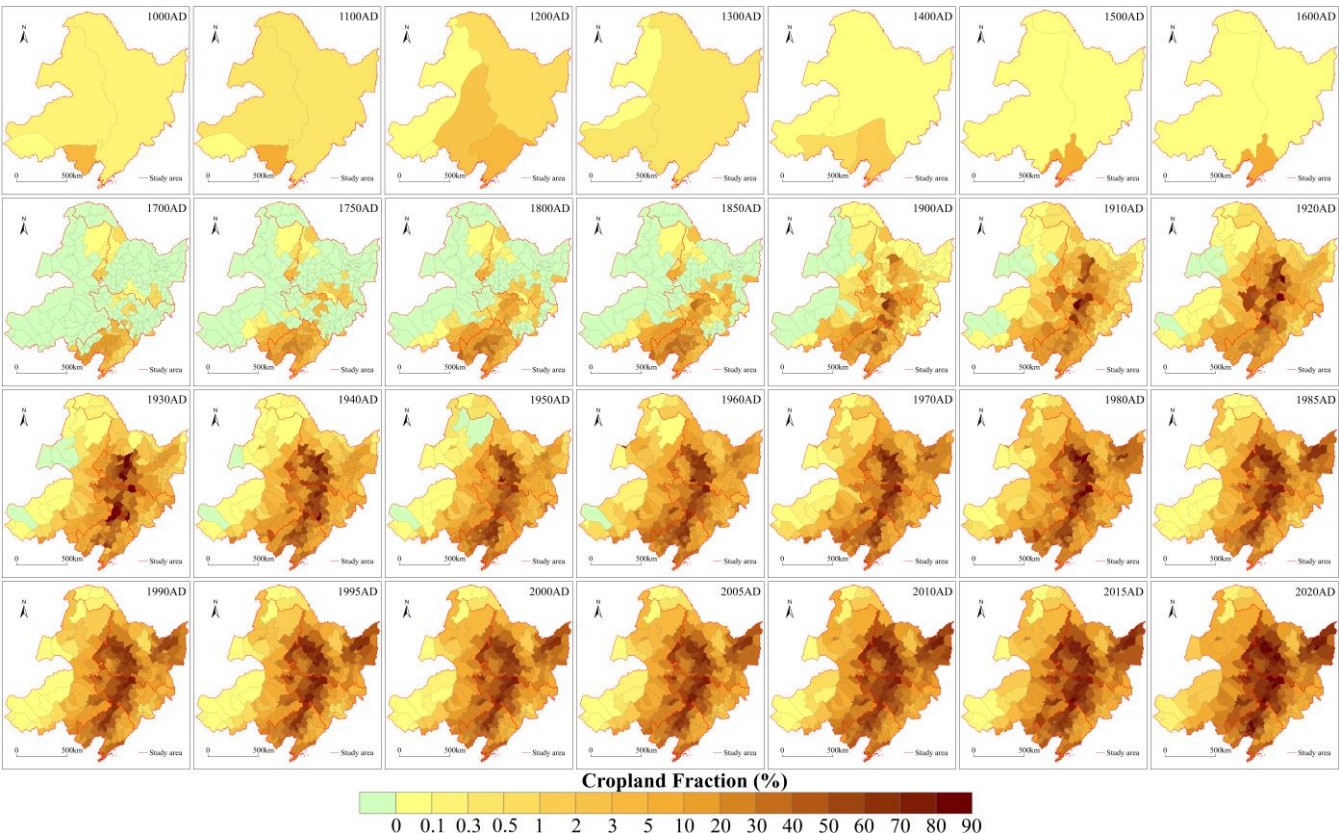

**Figure 5: Changes in spatial patterns of cropland in the Northeast China from 1000 to 2020.**

## 4 Discussion

### 4.1 Credibility assessment

Based on the study of Fang et al. (2020), three methods including accuracy assessment, rationality assessment, and likelihood assessment, can be used to assess the credibility of historical LUCC dataset. Regarding the likelihood assessment, in reconstructing cropland area from 1985 to 2020, we selected eight RS products to assess the consistency. Based on the control of cropland survey data, this study identified high-consistency and high-priority pixels as cropland pixels for this dataset and evaluated and validated the accuracy of the integration results. Theoretically, compared with any single RS products used in this study during this period, the total amount of cropland area in this study is relatively more accurate and the spatial distribution is relatively more reasonable.

### 4.1.1 Accuracy assessment

The cropland data at lower spatial scales can be used to evaluate the accuracy of reconstructed cropland area. Due to the availability of county-level cropland survey data, we selected the county-level first general land investigation at 1985 and the county-level data from the third NLS at 2020 for comparison. As shown in Fig. 6, the determination coefficients between the cropland area from this study and the cropland area from the survey data for 1985 and 2020 are 0.9582 and 0.9892 respectively.

This indicated that the overall accuracy of the reconstructed cropland area at county-level was relatively high, and our
constrained integration method that combines multisource cropland cover products with survey data can well match the spatial
distribution of cropland cover in Northeast China.
In addition, from 1985 to 2020, the identified high-consistency and high-priority pixels as cropland pixels based on
constrained integration method may lead to errors with survey data (Table S3). The relative errors between the cropland area
of this study and the cropland survey data for the period 1985 to 2020 as -1.35%, 4.02%, 5.17%, 1.10%, 0.21%, -1.93%, 0.25%
and 0.67%, respectively. The vast majority of errors are around 1%, with the larger errors are 4.02% and 5.17%, which indicates
that the reconstructed cropland area in this study is relatively accurate from 1985 to 2020.

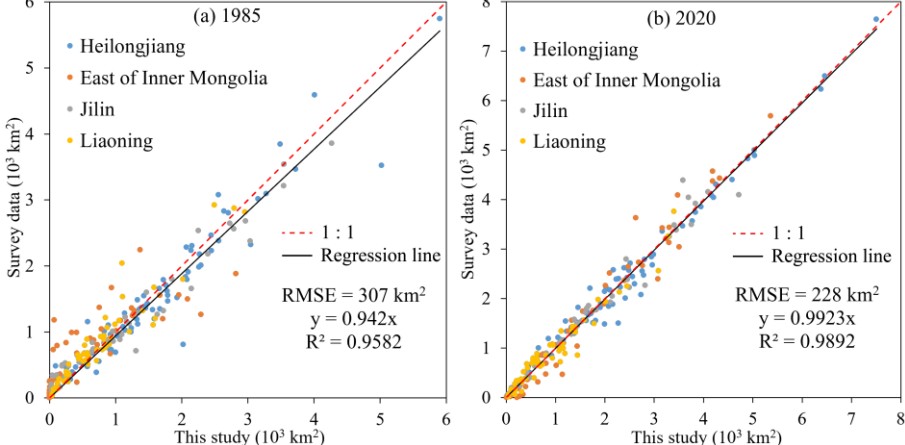

**Figure 6: Correlation between the cropland data of this study and survey cropland data at county-level in the Northeast China in**
**1985 and 2020.**

**4.1.2 Rationality assessment**
Due to the unavailability of actual historical land cover data, we used the actual historical agricultural development of Northeast
China as a reference standard for rationality assessment. As one of the cases evaluating the distribution rationality of the
HYDE3.2 cropland cover in Northeast China over the past millennium, Fang et al. (2020) analyzed changes in the northern
boundary and spatial distribution of settlement relics in the Liao, Jin, Yuan, and Ming periods (916~1644), as well as changes
in the cumulative number of towns and spatial distribution of towns in the three provinces of Northeast China during the Qing
Dynasty (1644~1911). The unique development history of the Northeast China shown in this case is basically consistent with
the process of increase or decrease and spatial distribution of the total cropland area during the same period reconstructed by
this study, which reflects the rationality of this dataset.
In addition, this study attempts to briefly summarize the population changes, settlements changes (the settlement relics
and the administrative division points derived from Jia et al. (2018) and the Historical Atlas of China (Tan, 1982a; Tan, 1982b))
(Fig. S2), warfare, and land policies that may have influenced land cultivation in Northeast China during the Liao, Jin, Yuan,
and Ming periods (1000~1600). The population and settlements in Northeast China from 1000 to 1600 exhibited phase changes

of expansion-reduction-expansion, with possible reasons including the Liao and Northern Song Dynasties signed the "*Chanyuan Alliance* (澶渊之盟)" in 1004 after war, the Jin and Southern Song Dynasties signed the "*Shaoxing Peace Treaty* (绍兴和议)" in 1141 after war, the Jin and Southern Song Dynasties signed the "*Longxing Peace Treaty* (隆兴和议)" in 1164 after war. During the three treaties and related wars, both the Liao and Jin dynasties in the north benefited significantly. They not only received reparations but also resettled large numbers of captives to the present-day Northeast China to engage in agricultural and other productive activities. Historical records also indicate that the rulers of the Liao and Jin dynasties during this period both attached much importance to agricultural production (Wu and Ge, 2022; Han, 1999; Toqto'A, 1974; Toqto'A, 1975).

From 1211, when Genghis Khan personally led the Mongol army to attack the Jin Dynasty, until 1233, the Mongols had essentially gained control over the entire Northeast China. Using this region as a base, they also conducted war against Goryeo (present-day Korean Peninsula), which lasted until 1259. From 1259 to 1287, the Mongols made several attempts to establish governing institutions in Northeast China, but faced continuous rebellions. It wasn't until the Yuan Dynasty subdued the rebellions and established the Liaoyang Province in 1287 that effective governance began in the Northeast China. However, during this period, the region suffered from continuous warfare, significant population loss, and severe disruptions to agricultural production (Xue, 2006, 2012). According to the Dynastic History of Yuan Dynasty, from 1294 to 1345, the Yuan government provided relief to Liaoyang Province 40 times. Additionally, rebellions in the Northeast China persisted from 1343 onwards, only being effectively subdued the rebellions by 1362, just six years before the collapse of the Yuan Dynasty in 1368 (Song, 1976; Xue, 2006, 2012).

In 1368, the Ming Dynasty was established, and remnants of the Yuan Dynasty retreated to the northern grassland, known as the Northern Yuan Dynasty (Tatar), which partly within our study area. It wasn't until 1389 that the Ming Dynasty established the "*Uriyangqa three Commanderies* (兀良哈三卫)" in the region from present-day Qiqihar city to Baicheng city, gaining certain practical control over the region. However, from 1399 to 1402, the Ming Dynasty faced the internal strife of the "*Jingnan Campaign* (靖难之役)" weakening its influence over the Northeast China, allowing some ethnic minorities to further occupy territories to the south. In 1409, the Ming Dynasty established the Dusi of Nuergan, reflecting their policy of appeasement and assimilation towards ethnic minorities in the Northeast China. In 1449, the Ming Dynasty experienced the "*Tumu Crisis* (土木之变)", prompting substantial efforts to fortify defensive structures. This also greatly strengthened the defensive capabilities of the Ming Great Wall in the Northeast China and confined the major agricultural population and agricultural areas of the Northeast China within the Dusi of Eastern Liao (south of the Ming Great Wall in the Northeast China). This situation persisted until the Ming Dynasty's collapse in 1644 (Cao and Ge, 2022; Fan, 2015; Cao and Ge, 2005; Zhang, 1974). All these pieces of evidence contribute to the validation of the rationality of our dataset to a certain degree.

**4.2 Comparison with global historical LUCC datasets and previous studies**

To better showcase the achievements of this study, we chose to compare our results with widely used global historical LUCC datasets (Fig. 7, Table S3): the History Database of the Global Environment (HYDE3.2) (Goldewijk et al., 2017), the Sustainability and the Global Environment (SAGE) (Ramankutty et al., 2008; Ramankutty and Foley, 1999), the Kaplan and Krumhardt 2010 (KK10) (Kaplan et al., 2011), and the Pongratz Julia (PJ) (Pongratz et al., 2008). Overall, the cropland area curve of Northeast China in this study is generally between the HYDE3.2 dataset and the PJ dataset. The SAGE dataset, KK10 dataset, and PJ dataset consistently show significantly higher values than the results of this study throughout the past millennium. It's worth noting that the KK10 dataset provides the combined area of cropland and pastureland, making it notably larger than the results of this study compared to other datasets. The SAGE dataset, which obtained cropland area data using an improved method in 2000, is relatively close to the results of this study. The curve of the PJ dataset is essentially consistent with the SAGE dataset from 1700 to 1990 because the cropland data in the PJ dataset during this period are derived from the SAGE dataset.

From the trend of the curve (Fig. 7), the HYDE3.2 dataset maintains a relatively low level of cropland area from 1000 to 1700. In comparison with this study, it fails to demonstrate the historical fact of cropland cultivation in the study area from 1000 to 1200. The HYDE3.2 dataset shows an increase in cropland area after 1700, with a growth rate similar to this study. The growth rate significantly rises after 1900, but during this period, its growth rate is notably lower than in this study. The SAGE dataset maintains a relatively high total cropland area and growth rate from 1700 to 1950. Subsequently, cropland area starts to decline, approaching the results of this study in the year 2000. However, the total cropland area in the SAGE dataset from 1700 to 2000 is significantly higher than the results of this study. The KK10 dataset exhibits drastic fluctuations from 1000 to 1850, with significant declines in the periods 1200 to 1300 and 1600 to 1700, placing the two points at the trough. For the remaining periods, it maintains a growing trend, and the total area of cropland and pastureland in the KK10 dataset from 1000 to 1850 is significantly higher than the cropland area in this study. The PJ dataset shows a fluctuating upward trend from 1000 to 1700, with trends in growth and decline generally consistent with this study during this period. The minimum cropland point is also around 1400, and after 1700, the total cropland area and growth rate in the PJ dataset are consistent with the SAGE dataset. The cropland area in the PJ dataset is significantly higher than this study from 1000 to 1990.

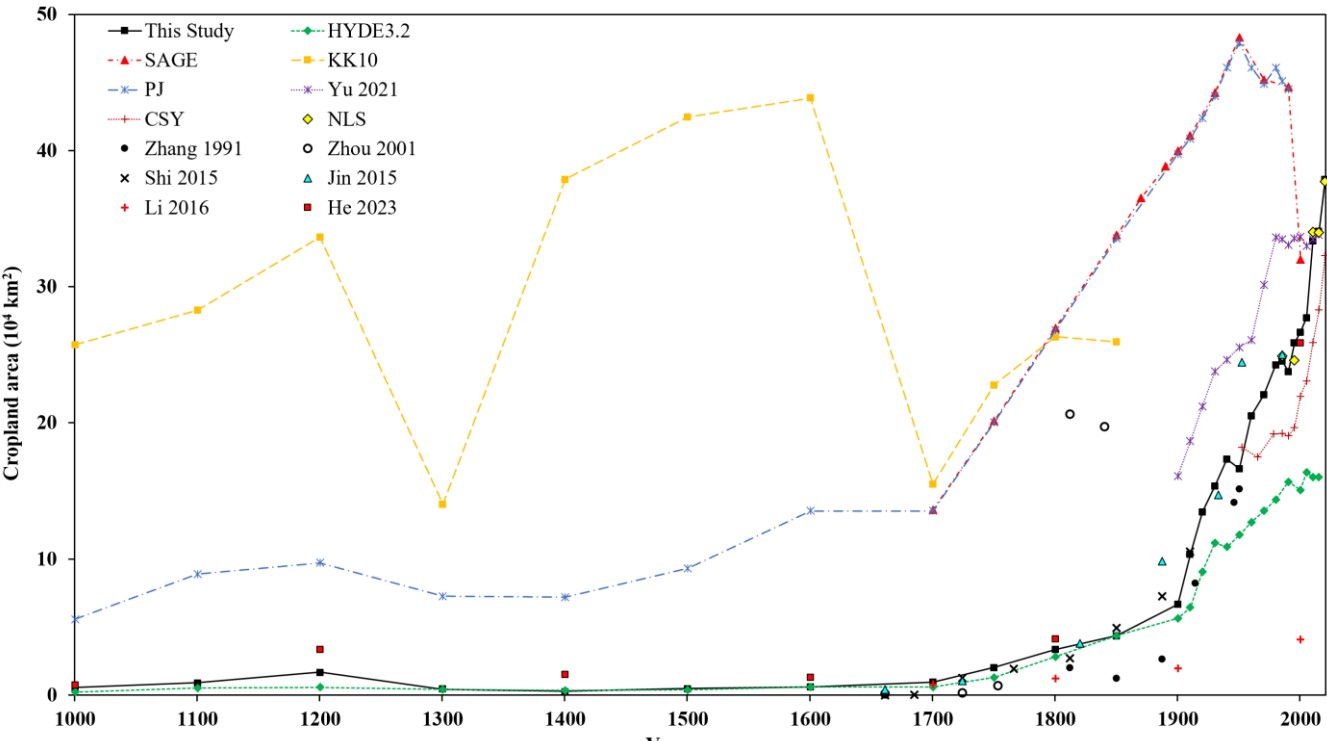


**Figure 7: Comparison of total cropland area from global historical LUCC datasets, previous studies and this study in the Northeast**
**China. The abbreviations used in the figure are as follows: HYDE3.2 refers to Goldewijk et al. (2017); SAGE refers to Ramankutty**
**et al. (2008) and Ramankutty and Foley. (1999); KK10 refers to Kaplan et al. (2011); PJ refers to Pongratz et al. (2008); Yu 2021**
**refers to Yu et al. (2021); CSY denotes the Chinese Statistical Yearbook (refer to provincial and prefectural statistical yearbook);**
**NLS denotes the National Land Survey (1985 refer to the first general land investigation; 1995 refers to the first national land survey;**
**2010 and 2015 refer to the second national land survey; 2020 refers to the third national land survey); Zhang 1991 refers to Zhang**
**(1991); Zhou 2001 refers to Zhou (2001); Shi 2015 refers to Shi (2015); Jin 2015 refers to Jin et al. (2015); Li 2016 refers to Li et al.**
**(2016); He 2023 refers to He et al. (2023).**

This study was compared the total cropland area with previous representative published studies in Northeast China (Fig.
7, Table S3).
The data from Shi (2015) for 1661 and 1685 are significantly lower than this study, at these two time-points, Shi (2015)
only had the data from Fengtian (roughly equivalent to Liaoning Province). Although the data from Shi (2015) for 1724
included the total area for Heilongjiang, Jilin, and Fengtian, the territorial scope of Heilongjiang and Jilin during this period
was larger than that of present-day Heilongjiang and Jilin provinces. We did not exclude the cropland area according to the
proportion of these territory outside present-day China. Additionally, there were 15.35, 15.35, and 17.35 million Qing *Mu*
(9431 km$^2$, 9431 km$^2$, and 10660 km$^2$) of banner cropland at these three time-points, mainly distributed in Zhili (partly within
our study area) and various parts of Northeast China, which could not be accurately divided. Therefore, we did not include the
banner cropland for these three time-points. For the Mongolia in 1766, 1812, 1850, 1887, and 1911 from Shi (2015), we
converted the data based on the area proportion of the Qing Dynasty Mongolia within our study area, which is 41.58%.
The data from Jin et al. (2015) closely matches this study growth trend. Both studies acknowledge that the 1985 land

survey data is relatively accurate, resulting in no significant differences at this point. In Jin et al. (2015), the data in the Inner Mongolia in 1661 is missing, and for subsequent time-points, we calculated the data based on the area proportion of the East of Inner Mongolia (within our study area), which is 55.26%. The data in 1661, 1724, 1820, 1887, 1933, and 1952 from Jin et al. (2015) is similar to this study. The main reason for the difference in cropland area between the two studies may be due to the specific data source and data adjustment methods.

The data from He et al. (2023) closely matches this study growth trend. It should be noted that, for clearer comparison with our study, we selected standard time-points every 200 years from 1000 to 2000 on the cropland area curve from He et al. (2023). Similarly, we calculated the cropland area in the East of Inner Mongolia (within our study area) based on the proportion of 55.26%. In He et al. (2023), the data from 1000 to 1800 is slightly higher than this study, possibly because of the different methods for reconstruct the cropland area based on the population and the different proxy indicators used by the two studies during this period.

Similar to the comparison with He et al. (2023), when selecting the CHCD data from Li et al. (2016) for comparison, we chose standard time-points every 100 years from 1700 to 2000 on the cropland area curve for Inner Mongolia from CHCD data, and calculated the area for the East of Inner Mongolia based on 55.26%. The CHCD data for Heilongjiang, Jilin, and Liaoning province is consistent with our study (Ye et al., 2009), however, this study corrected Ye's data as explained earlier (Table S1). The difference in cropland area for the East of Inner Mongolia between the two studies may be due to the calculation of cropland area based on the proportion of 55.26%, which may not align with the actual historical agricultural development of Inner Mongolia.

For the sake of clear comparison, we selected standard time points every decade from 1900 to 1980, and every five years from 1985 to 2015 from Yu et al. (2021). The difference between the two studies in 2015 is minimal, as both studies acknowledge the NLS data is relatively accurate. The main reason for the difference in cropland area between the two studies may be due to the different reconstruction methods. In Yu et al. (2021), the officially released NLS data in 2017 (cropland area in 2016) is used as the benchmark data, and an assumption was made that this most recent data is the most reliable. Then calculated the national cropland area by using the NLS data in 2017 as the baseline and adjusted using the interannual variation information of cropland derived from various sources. Due to the cropland area difference between the national total and the sum of the provincial, a proportional adjustment was applied to match the provincial to the national total. In this study, three times NLS data were adopted, we assumed that the cropland survey data is the most reliable, then corrected the statistical data through the correlation coefficients between the statistical and the survey data in different periods to obtain the cropland area for the time-points without survey data.

The data from Zhang (1991) consistently shows lower values compared to this study across all time points. The differences may arise because the lack of data in Inner Mongolia for all periods except 1949 from Zhang (1991). Both studies agree that

national statistical data is reliable for 1950s, the data from Zhang (1991) is slightly underestimates compared to our study,
likely due to the calculation of cropland area based on the proportion of 55.26% in the East of Inner Mongolia.
The data from Zhou (2001) shows lower values compares to this study in 1661, 1724, and 1753. The differences may
arise because the lack of data in Heilongjiang, Jilin province, and Inner Mongolia in these periods. Conversely, in 1812 and
1840, the data from Zhou (2001) is significantly exceeds to our study. The differences may arise because the assumption of
the cropland. In Zhou (2001), an important assumption is that the northern territorial boundaries were much larger than today,
then the cropland area of Heilongjiang, Jilin, and Liaoning province in 1952 were used instead of the cropland area in 1812
and 1840. This assumption may contradict the actual historical agricultural development of Northeast China.
**4.3 Spatial distribution of cropland cover compared with HYDE3.2 dataset**
We acknowledged that there is no more credible cropland area data at the global scale than HYDE up to now. Compared to
this study, the HYDE3.2 dataset shows relative differences ratio (RD) in total cropland area for the period 1000 to 1600 as -
82.92%, -52.52%, -100.45%, -5.32%, 17.42%, -29.34%, and 0.55%, respectively (Fig. 8~9). The relative differences ratio (RD)
as shown in Equation (4):
$\mathrm{RD} = \frac{C_\mathrm{H}(y) - C_\mathrm{T}(y)}{(C_\mathrm{H}(y) + C_\mathrm{T}(y))/2} \times 100\%$ ,                                                              (4)
where $C_\mathrm{H}(y)$ represents the total cropland area from HYDE3.2 for year $y$, and $C_\mathrm{T}(y)$ represents the total cropland area from
this study for year $y$.
Compared to this study, except for the years 1100 and 1300, where the absolute values of RD in most provinces within
the study area did not exceed 50%, for other years, most provinces showed relatively large RD. In the years 1000 and 1100,
except for certain areas in Xilin Gol League where the HYDE3.2 dataset showed more cropland area, the rest of the regions
generally had less cropland area than this study. In 1200, the HYDE3.2 dataset showed more cropland area in the western
region, while the opposite was observed in the eastern region. In 1300, the HYDE3.2 dataset indicated less cropland area in
the entire region. From 1400 to 1600, the HYDE3.2 dataset showed more cropland area in the northern region. As the scope
of the Dusi of Eastern Liao reduced, this study's cropland area in this region significantly exceeded the HYDE3.2 dataset. In
1700, both the HYDE3.2 dataset and this study indicated that most counties in Heilongjiang and Jilin provinces, as well as the
northeastern part of Inner Mongolia, had no cropland (Fig. 5, Fig. 8). However, the HYDE3.2 dataset showed that during this
period, a considerable area of cropland existed in most regions of Inner Mongolia and the Sanjiang Plain, leading to 34.38%
of county-level RDs being greater than 100% (Fig. 9). From 1750 to 1850, the HYDE3.2 dataset showed that the expansion
of cropland cultivation gradually extended northward to cover the entire region (Fig. 8). This contradicts the areas without
cropland caused by the abandoning reclamation restrictive policies of the Qing government during this period. Additionally,
during this period, in the counties which both datasets considered with cropland, this study found that, except for a few counties
where cropland area was less than the HYDE3.2 dataset, most counties had significantly more cropland area in this study.
During this period, over half of the counties in the study area had RDs greater than 100%. From 1900 to 1950, as the abandoning
reclamation restrictive policies, this study observed a decreasing trend in cropland fraction from the center to the periphery in
the study area (Fig. 5). Compared to the HYDE3.2 dataset, counties with RD greater than 100% gradually decreased (Fig. 9).
Furthermore, during this period, in most areas of the Songnen Plain and the Liaohe Plain, this study's cropland area was
significantly greater than the HYDE3.2 dataset. After 1950, the RD for each county in the study area gradually decreased and
concentrated in the (-100%, -10] range (Fig. 9), indicating that the cropland area in most counties in this study was significantly
greater than the HYDE3.2 dataset.

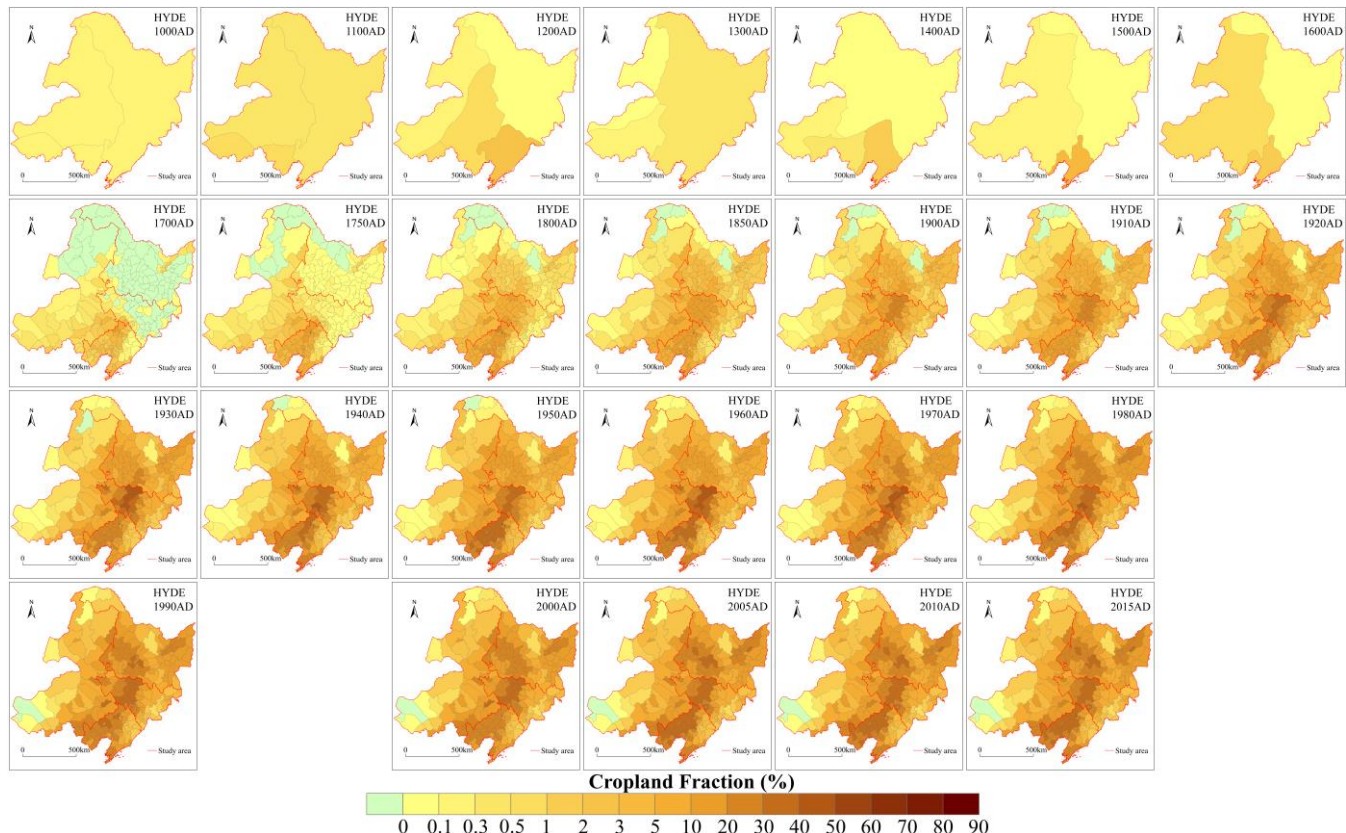

**Figure 8: Changes in spatial patterns of cropland of HYDE3.2 dataset in the Northeast China from 1000 to 2015.**

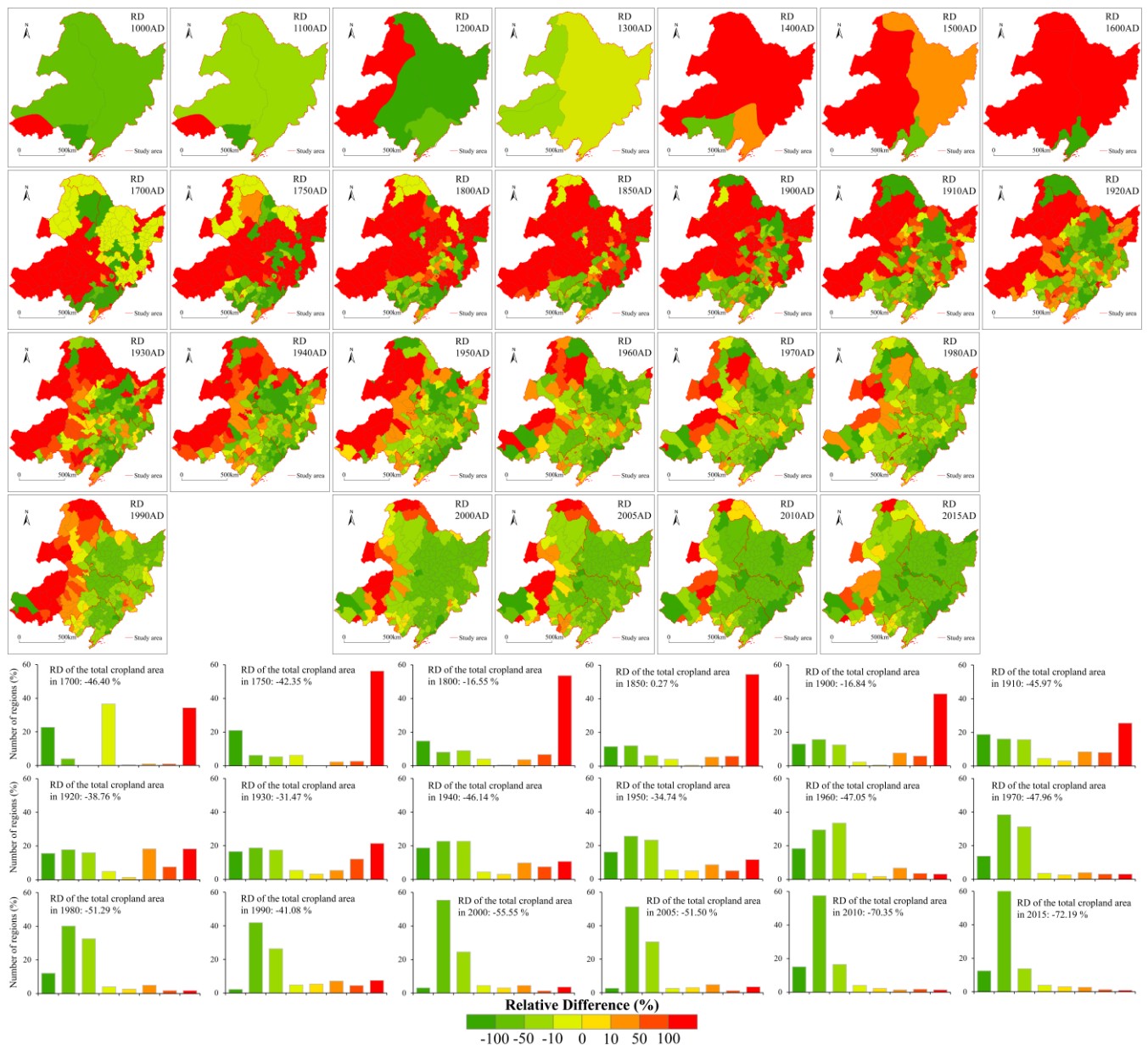

**Figure 9: Comparison of the spatial distribution of cropland area between HYDE3.2 and this study in the Northeast China.**

**4.4 Uncertainty analysis**

In this study, the uncertainty mainly consisted in two aspects: the definition and selection of data, the application of methods. Regarding the data aspect: (1) In this study, the definition of cropland before 1950 is: the sum of arable land and land under permanent crops, and the temporary changes in land use and fallow land during historical periods were not considered. The cropland area for 1950 and later are basically consistent with the identification rules in the NLS. Although the temporary changes in land use and fallow land during historical periods, this may still result in our reconstruction slightly less cropland than actual historical period.

   (2) Due to the completeness of historical documents, the reconstruction results of cropland for seven time points from 1000 to 1600 in this study are at the provincial-level, which may not finely reflect the spatiotemporal characteristics of cropland.

Especially between 1000 and 1300, the results may lead readers to mistakenly believe that cropland were evenly distributed across the entire Northeast China. However, based on the distribution of settlement relics during this period, cropland may mainly distribute on the Liaohe Plain and on the southern part of the Songnen Plain, then reduced southward into Liaoning Province.

(3) The two proxy indicators of 14 *Mu* (0.93 hm$^2$, the average annual potential cropland area per Man of the agricultural population) and 2 *Mu* (0.13 hm$^2$, the average cropland area per household in the nonagricultural population) from 1000 to 1600 may lead to inaccuracies in cropland estimation. The reasons for using population to reconstruct cropland during this period have been detailed in the previous section, necessitating further analysis and clarification of the corresponding cropland-related indicators.

Firstly, the conclusion of 14 *Mu* per Man for agricultural population during the Liao and Jin Dynasties (1000~1200) is primarily derived from historical records in the Jin Dynasty (1200) and the relationship between population and cropland in the early Qing Dynasty (1661~1680) (Jia et al., 2023). There are two reasons why 14 *Mu* was used in the Yuan and Ming Dynasties (1300~1600): one reason is the agricultural household size and the ratios of Man in agricultural household in Northeast China during the Yuan and Ming Dynasties (1300~1600) are closer to those of the Liao and Jin Dynasties (1000~1200) (Table 1). And the per capita cropland area owned by agricultural population in the Liao-Jin-Yuan-Ming periods (1000~1600) consistently ranged between 4 and 5 *Mu* (0.27~0.33 hm$^2$), slightly higher than the subsistence level of 3 *Mu* per capita in previous studies for the same historical period in this region (Ye et al., 2009; Fang et al., 2006; Shi, 1990), which is relatively reasonable. The second reason is that there were no significant changes in agricultural production technology in Northeast China during the Liao-Jin-Yuan-Ming periods (1000~1600), and the population declined significantly compared with the Liao and Jin Dynasties (1000~1200) due to factors such as warfare. However, considering the social stability at standard time-points during the Yuan and Ming Dynasties (1300~1600), the strong willingness of the agricultural population towards cultivation, and the limitations of individual cultivation capabilities, the cropland from the Liao and Jin Dynasties could be relatively easily inherited and reclaimed by descendants.

Secondly, Similar to the agricultural population, considering the non-agricultural household size, stable agricultural production technology, the historical inheritance of most ethnic groups, this study continues to use 2 *Mu* as the calculation indicator of non-agricultural population in the Yuan and Ming Dynasties (1300~1600) (Cong, 1993a; Cong, 1993b; Wu and Ge, 2005a; Cao and Ge, 2005b; Liu et al., 2016).

Regarding the method aspect: (1) From 1700 to 1980, cropland areas at multiple time points in this study were derived through linear interpolation and polynomial curve fitting. Although we have fully considered historical facts and other research conclusions (Fang et al., 2020; Ye et al., 2009; Fang et al., 2005) when selecting the interpolation time points, 1860 was chosen as the dividing point between slow growth and rapid growth. This method, compared to data recorded at each specific historical

point, may affect the accuracy of the value at those standard time points.

(2) From 1700 to 1980, the county-level administrative boundaries in the published data used in this study differ from the

modern county-level administrative boundaries used in this study. Especially in the CNEC data (Ye et al., 2009) in 1683, 1735
and 1780, there is county-level in Liaoning province, Assistant Governorate Jurisdiction (prefecture-level) in Heilongjiang and
Jilin province. This would result in counties belonging to different Assistant Governorate Jurisdictions in present-day having
the same cropland fraction. This problem is difficult to correct further because the lowest administrative level in Northeast
China available in historical data during this period is Assistant Governorate Jurisdiction (prefecture-level).

(3) From 1985 to 2020, the land survey data utilized in this dataset might exhibit uncertainties in early cropland data due

to backward technology and other factors. Additionally, the use of a uniform correlation coefficient to correct the cropland
statistics data for the entire Northeast China may affect the accuracy of the cropland area in localized areas, this may lead to a
lower cropland area at previous time-points. To mitigate the impact of these uncertainties on our dataset during this period,
this study mainly adopts two methods: Firstly, this study mainly selects the standard time-points data after the nationwide
surveys, avoiding the use of annual land change survey data. For instance, the cropland area in 1985 in this dataset is based on
the first general land investigation around 1985; the cropland area in 1995 is based on the first NLS's standard time-point data
on October 31, 1996; the cropland area in 2010 is based on the second NLS's standard time-point data on December 31, 2009;
and the cropland area in 2020 is based on the third NLS's standard time-point data on December 31, 2019. Secondly, this study
uses correlation coefficients to correct the statistical data by category and time point. For instance, the average correlation
coefficient of the second and third NLSs with corresponding statistical data is used to correct the statistical data for the 1990,
2000, and 2005; the correlation coefficient of the 1985 first general land investigation with corresponding statistical data is
used to correct the statistical data for the 1950~1980.

**5 Data availability**

All cropland data reconstructed in this study are publicly available at https://doi.org/10.6084/m9.figshare.25450468.v2 (Jia,

2024).


**6 Conclusion**

Based on historical documents, proxy data such as population data, revised published results, remote sensing data products,
statistical data, and survey data, and utilizing a series of data processing methods, as well as accuracy and rationality assessment
methods, we established a 28 time-points cropland area dataset in Northeast China at provincial-level and county-level spatial
resolutions from 1000 to 2020. Reconstruction results indicate that cropland area in Northeast China grew slowly before 1850
and experienced rapid expansion after 1850, maintaining this growth trend until 2020. This dataset illustrates the characteristics
of cropland changes in Northeast China over the past millennium, especially in the past 300 years. Between 1000 and 1200,
the extent of cropland was roughly equivalent to the modern era. Subsequently, until 1850, the cropland was mainly
concentrated in the Liaoning Province. However, with the Qing government establishing military garrisons in the northern part
of the Northeast China, farming areas was formed around these garrisons from 1700 to 1850. With the implementation of the
immigration and cultivation policy in the latter half of the 19th century, the spatial pattern of cropland coverage in Northeast
China changed significantly after 1850, with agricultural zones rapidly expanding across the entire region. After 1950, the
expansion of high cropland fraction agricultural zones in Northeast China became more pronounced, gradually forming core
areas with high cropland fraction in the Liaohe Plain, Songnen Plain, and Sanjiang Plain.
Despite the fact that the cropland area change dataset in this study is presented at the provincial-level and county-level,
the dataset we reconstructed based on historical records at 28 time points can be approximated as "truth value". This dataset
provides crucial support for the long-term land use changes in the Northeast China. In the future, we will further investigate
gridded cropland allocation methods based on the historical cultivation process in the Northeast China, aiming to better serve
research such as carbon emission, climate data construction, climate-ecosystem modeling and the conservation and utilization
of black soil, etc.

**Author contributions.** RJ, XF and Yu Y designed this work. RJ wrote the manuscript. XF and Yu Y provided suggestions on
methodology. Yu Y and Yundi Y developed the dataset. All the authors contributed to the review of the manuscript.

**Competing interests.** The authors declare that they have no conflict of interest.

**Financial support.** This research has been supported by the National Key R&D Program of China (grant no.
2021YFD1500704), and the Fund of the China Scholarship Council (grant no. 202306040062).

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
