# Peer review of "A 28 time-points cropland area change dataset in Northeast China"

_Earth System Science Data, 2024_

## Author Comment (AC1)

**Response to referee comments**

We thank the two reviewers and the editor for the precious and constructive suggestions to improve our manuscript. We carefully revised our manuscript and addressed the comments of each of the two reviewers. Please find our point-by-point response below.

**Referee #1**

General comments. According to the World Meteorological Organization, 2023 is the hottest year on record. Therefore, it is of great significance to develop a long-term cropland dataset to explore the climatic effects of human land use. This study reconstructs millennial cropland for Northeast China. Topics fits the aims and scope of the ESSD. The following comments and suggestions should be considered for revisions.

Response: We sincerely thank the reviewers for valuing our work. Thank you for your insightful comments which have improved our work greatly. Please find our point-by-point response below.

1. --First, why only area estimation, and no spatial reconstruction? A 5' × 5' cropland dataset is developed for Northeast China from AD 1000 to 1200 by these authors (Gridded reconstruction of cropland cover changes in Northeast China from AD 1000 to 1200. https://link.springer.com/article/10.1007/s10113-023-02118-y). But in this study, only provincial-level or county-level cropland area is available. Why? Obviously, the datasets reconstructed in this study cannot be used by climate modelers. In addition, in terms of data from 1000 to 1200 years, is there any improvement in this paper compared to the paper mentioned above (Gridded reconstruction of cropland cover changes in Northeast China from AD 1000 to 1200)?

Response: Thank you for your helpful suggestion. In this study, 1000 to 1600 corresponds to historical provincial-level administrative districts, while 1700 to 2020 corresponds to modern county-level administrative districts.

The primary reasons are as follows: First, compared to the regional existing historical LUCC gridded reconstruction results, the cropland area data of administrative

units is relatively more fundamental and reliable. Our research experiments indicate that there remains a certain degree of uncertainty in gridded reconstruction of cropland over long historical periods, even when using human factors supported by historical data in the allocation model (such as historical settlement points) (Jia et al., 2023). Long-term historical cropland gridded reconstructions need to consider using allocation methods that match the historical facts of different periods in local area. In theory, based on this dataset, researcher could even create datasets with a resolution of 1 meter or less. If climate modelers need to use gridded cropland datasets, they can easily convert our dataset into a customized grid spatial resolution dataset according to their required time range and main influencing factors (such as natural factors, human factors, etc.). For instance, some studies concluded that the HYDE dataset can be used as a map of agricultural potential and crop suitability, especially in periods before the advent of satellite imagery (Yu and Lu, 2018; Yu et al., 2021).

Second, for the 1700 to 2020 corresponds to modern county-level administrative districts, the average spatial scale of these counties in this dataset ranges between 0.5°×0.5° and 1°×1°. Theoretically, when readers use this dataset to convert it into the gridded dataset they need, the error can be controlled within 1°×1° even if they don't use natural or human factors to guide the allocation.

Third, this dataset can be applied in a wide range of scenarios (such as carbon emission and carbon neutrality, climate data construction, ecological footprint, and biological population assessment, etc.). The bilingual format and the administrative boundaries consistent with the current county-level administrative units in China also facilitate its use by scholars in the humanities and social sciences worldwide.

Realistically, compared to the primary data and reconstruction methods of Jia et al. (2023), this study directly used the results of the cropland area (1000-1200) of the above study. The main difference is that, this dataset provides provincial-level cropland area data for three time points (1000-1200) within the current administrative boundaries of Northeast China, consistent with the boundaries of the other 25 time points in this dataset. We are also very pleased to offer the reconstructed gridded cropland dataset in Northeast China from 1000 to 1200 as a reference solution for readers.

References:

Yu, Z. and Lu, C.: Historical cropland expansion and abandonment in the continental U.S. During 1850 to 2016, Glob. Ecol. Biogeogr., 27, 322-333, https://doi.org/10.1111/geb.12697, 2018.

Yu, Z., Jin, X., Miao, L., and Yang, X.: A historical reconstruction of cropland in china from 1900 to 2016, Earth Syst. Sci. Data, 13, 3203-3218, https://doi.org/10.5194/essd-13-3203-2021, 2021.

Jia, R., Fang, X., and Ye, Y.: Gridded reconstruction of cropland cover changes in Northeast China from ad 1000 to 1200, Reg. Envir. Chang., 23, 128, https://doi.org/10.1007/s10113-023-02118-y, 2023.

2. --Second, the applicability of the reconstruction method of estimating the cropland area for a small area by population. Generally speaking, estimating cropland by population is mostly applicable at continental to global scales. In the case of a small region, more other factors will affect the relationship between population and cropland. Response: Thank you for your suggestion. The fundamental reason is that there are no direct historical records of cropland area in Northeast China from 1000 to 1600. When we selecting proxy indicators to reconstruct cropland area, the population data for this period is relatively complete and authoritative.

We acknowledge that at a regional scale, the factors affecting the relationship between population and cropland area are diverse and may change over time. Therefore, we combine historical facts (particularly from 1000 to 1600) and use the most authoritative historical population data in China: "*The History of Population of China*" (Wu and Ge, 2005a; Cao and Ge, 2005b). Based on the needs of this study, we extracted the population data (mainly divided into agricultural, non-agricultural, and military populations) that was consistent with the scope of this study area, and developed cropland calculation indicators for different historical periods corresponding to different population categories.

In several global LUCC datasets, such as HYDE, when estimating cropland based

on population, either the population indicators (e.g.: per capita cropland) remain unchanged across different historical periods; or in the uncertainty estimates, homogeneously varied it with time across the globe; or to account for their uncertainties, different population databases were used and the upper and lower ends of an uncertainty range were assessed. However, these estimation methods can still lead to errors in cropland area on a global or regional scale. Therefore, when this study used historical population data to reconstruct cropland, special attention was paid to two key points: 1) authoritative and accurate population data, and 2) cropland areas corresponding to different population categories based on historical facts (Please see Line 592-614). Thus, we believe this method is relatively applicable for Northeast China from 1000 to 1600.

References:

Wu, S. and Ge, J.: The History of Chinese Population, Volume 3, Fudan University Press, Shanghai, China, 2005a (in Chinese).

Cao, S. and Ge, J.: The History of Chinese Population, Volume 4, Fudan University Press, Shanghai, China, 2005b (in Chinese).

3. --Third, failure to evaluate the reliability or accuracy or uncertainties of the reconstructed dataset will affect the user's use of the dataset. The comparison with the global dataset does not indicate the reliability of the dataset developed in this paper, because the global dataset itself has a large degree of uncertainty. The fact that the reconstruction results in this paper are very different from the global dataset does not mean that the dataset developed in this paper is reliable.

Response: Thank you for your insightful suggestion. We acknowledge that the current paper's reliability, accuracy, or uncertainties assessments are not yet sufficiently comprehensive. We have made every effort to supplement the relevant assessments and uncertainty analysis as much as possible. Please see Line 416-625, new Fig. 7 and Table S3.

Here is the new Fig. 7:

[Figure]

Figure 7: Comparison of total cropland area from global historical LUCC datasets, previous studies and this study in the Northeast China. CSY denotes the Chinese Statistical Yearbook; NLS denotes the National Land Survey.

4. --Fourth, writing is not done from the perspective of data development (Data description paper), it is more like a research paper. For example, the core content of the results should not be the analysis of the spatio-temporal characteristics of cropland changes, but the rationality, reliability, accuracy, and potential uses of the data products developed in this paper. More specific comments are as follows.

Response: Thank you for your insightful suggestion. We reorganized the paper to make it more like a data description paper. It mainly includes: 1. Deletion of the conclusion section (Please see Line 630-649); 2. Reorganized of introduction, results and discussion sections (Please see Line 26-74, 382-578); 3. Add the credibility assessment and uncertainty analysis (Please see Line 416-625); 4. Potential uses are added to the introduction and abstract sections (Please see Line 22-24, 38-40).

5. --Title and Introduction. Why reconstruction for 1000 to 2020 in Northeast China? More explanations are necessary. Based on Figure 5, From 1000-1700, there was only a small area of cropland in Northeast China. Line 393, In 1200, cropland fraction of

1.17%; In 1400, line 395, cropland fraction of only 0.19%. The environmental impact of such a small area of cropland is completely negligible. Based on figure 5, the topic for past 300 years (Ye, Y., Fang, X., Ren, Y., Zhang, X., and Chen, L.: Cropland cover change in northeast china during the past 300 years, Science China Earth Sciences, 52, 1172-1182, https://doi.org/10.1007/s11430-009-0118-8, 2009.) is good, but for 1000 to 2020 may be not a good research topic.

Response: Thank you for your insightful suggestion. We chose to reconstruct the cropland changes in Northeast China over the past millennium mainly for the following reasons: 1. From the LUCC projects carried out by IGBP and IHDP in the last century to the recent PAGES (The PAGES (Past Global Changes) project is an international effort to coordinate and promote past global change research. The primary objective is to improve our understanding of past changes in the Earth system in order to improve projections of future climate and environment, and inform strategies for sustainability.), LandCover6k (The goal of LandCover6k was to produce datasets on past land-cover and land-use on continental and global spatial scales that are useful for climate modeling studies on land-use as a climate forcing.) and other research projects, the importance of accurate long-term historical LUCC datasets has been emphasized.

2. The land reclamation in Northeast China exhibited a unique pattern: during the period between the two land reclamations (eleventh and twelfth centuries; from the nineteenth century to present), there was a prolonged period of nomadism in this area (Jia et al., 2023). In addition, the global historical LUCC datasets fail to demonstrate the historical fact of cropland cultivation in the study area from 1000 to 1200.

3. We want to better demonstrate the changes in human impact on terrestrial environments, ranging from near-natural original states (natural dominance) to significant alterations induced by extensive human intervention (anthropogenic dominance), particularly since the Industrial Revolution.

4. Northeast China has now become one of the most important agricultural regions in China and the world. Focusing solely on the past 300 years of research is not conductive to the exploration of long-term LUCC effects in critical agricultural areas worldwide (He et al., 2023). If feasible, we are willing to expand the dataset regarded

as "truth values" to encompass a broader historical span, thereby enhancing the credibility of historical cropland area.

References:

He, F., Yang, F., Zhao, C., Li, S., and Li, M.: Spatially explicit reconstruction of cropland cover for china over the past millennium., Science China Earth Sciences, 66, 111-128, https://doi.org/10.1007/s11430-021-9988-5, 2023.

Jia, R., Fang, X., and Ye, Y.: Gridded reconstruction of cropland cover changes in Northeast China from ad 1000 to 1200, Reg. Envir. Chang., 23, 128, https://doi.org/10.1007/s10113-023-02118-y, 2023.

6. --Data and Methods. Not clear enough. For example, Line 91-115, It only introduces population data, per household population data, and interpolates the population according to the population growth rate, and does not involve how to estimate the cropland area at all. Line 107-110 mentions how to estimate the area of cropland, but it is very simple and there is no specific method. As far as Northeast China is concerned, why such an estimate is reasonable is not explained at all. From the perspective of historical land use reconstruction, estimating cropland area based on population as a proxy is only applicable to large-scale scales such as global and continental. For example, HYDE uses population to estimate the world's historical cropland. It also makes sense to reconstruct China's historical cropland in this way. But the Northeast is only a small region of China, so there is a lot of uncertainty in the results of this estimate.

Response: Thank you for your constructive suggestion, and we apologize if this was confusing. The main algorithm applied in the Liao, Jin, Yuan, and Ming Dynasties (1000-1600) can be found in the supplementary materials.

The reason and the credibility of using population to reconstruct cropland is similar to your second question, to which we have already responded and discussed more fully in the "Uncertainty analysis section". Please see Line 592-614. Regarding the importance of Northeast China, we added content in the "Introduction section". Please see Line 51-74.

7. --In addition, for 1000, 1100, and 1200, what's the difference between this study and the paper mentioned above (Gridded reconstruction of cropland cover changes in Northeast China from AD 1000 to 1200).

Response: Thank you for your comment. This comment is similar to your first question. Realistically, compared to the primary data and reconstruction methods of Jia et al. (2023), this study directly used the results of the cropland area (1000-1200) of the above study. The main difference is that, this dataset provides provincial-level cropland area data for three time points (1000-1200) within the current administrative boundaries of Northeast China, consistent with the boundaries of the other 25 time points in this dataset. We are also very pleased to offer the reconstructed gridded cropland dataset in Northeast China from 1000 to 1200 as a reference solution for readers.

References:

Jia, R., Fang, X., and Ye, Y.: Gridded reconstruction of cropland cover changes in Northeast China from ad 1000 to 1200, Reg. Envir. Chang., 23, 128, https://doi.org/10.1007/s10113-023-02118-y, 2023.

8. --Line 170-197;Line 257-278. Introduce too much about the estimation methods in published papers (Ye et al, 2009; Tian et al., 2005). It needs to be drastically cut, and readers can read these papers at all. In short, the writing of the method section is too lengthy and will scare off the vast majority of readers.

Response: Thank you for your helpful suggestion, and we apologize if this was confusing. We have deleted the detailed description about the estimation methods in published papers in the main text to make the article more readable. Considering that the papers we cited in this section is not available in English, we have added these contents to the supplement material for readers who need it.

9. --Line 232. Correct negative or zero values of cropland. If the estimated results have

a negative value, then there must be a problem with the previous interpolation and fitting methods, and we have reason to suspect that all the results obtained by the interpolation are problematic. Just correct negative or zero values of cropland isn't enough, what about the other results? From this point, it can be seen that this paper needs to have an uncertainty assessment of the estimation results, otherwise readers will not dare to use this data product to carry out downstream research.

Response: Thank you for your insightful suggestion, and we apologize if this was confusing. We have reorganized the structure of the original text, and added more specific and detailed description based on careful checking and correction of the existing errors in the original text. Please see Line 211-261 and the Table S2.

10. --Results. ESSD readers are more concerned about the reliability, availability, and accuracy of data products. However, the spatiotemporal variation characteristics of cropland area are not the most important.

Response: Thank you for your helpful suggestion. We acknowledge that the current paper's reliability, accuracy, or uncertainties assessments are not yet sufficiently comprehensive. We have made every effort to supplement the relevant assessments and uncertainty analysis as much as possible. And we deleted some description of the spatiotemporal variation characteristics of cropland area. Please see Line 382-412, 416-625.

11. --4.1 comparison. The comparison with the global dataset does not indicate the reliability of the dataset developed in this paper, because the global dataset itself has a large degree of uncertainty. The fact that the reconstruction results in this paper are very different from the global dataset does not mean that the dataset developed in this paper is reliable. Line 516-517, the following statement is not acceptable "Comparative analysis with global historical LUCC datasets indicates that the results of this study are relatively credible and more rational."

Response: Thank you for your insightful suggestion. "Comparative analysis with global historical LUCC datasets indicates that the results of this study are relatively credible

and more rational." This statement is indeed inappropriate. We have deleted such statements in the original text and used more objective language to describe the differences between different datasets and analyze the possible reasons for the differences. Please see Line 416-625.

12. -- technical corrections. Figure 5, no titles for x and y axes.

Response: Thank you for your suggestion. We revised the Fig. 5 and Fig. 7.

---

## Author Comment (AC2)

**Response to referee comments**

We thank the two reviewers and the editor for the precious and constructive suggestions to improve our manuscript. We carefully revised our manuscript and addressed the comments of each of the two reviewers. Please find our point-by-point response below.

**Referee #2**

The authors developed a dataset to document cropland area over the past 1000 years in the North China. By using historical records and recent datasets, the manuscript particularly looked at the spatial changes and possible improvement to the accuracy of the regional dataset. I have a few concerns and suggestions for the authors to consider if they decide to revise the manuscript.

Response: Thank you very much for your constructive comments on our manuscript. We appreciate the time you spent reviewing our manuscript. Please find our point-by-point response below.

1. The novelty of this manuscript is not clearly presented. The authors have already published a few similar papers in the past few years, and even one for the Northeast China region. The only difference is the time period covered here. Land use change, especially for such long history with spatial coverage, is deemed important in understanding carbon budget, land emissions, and many other studies. This is what the authors also emphasized in the Introduction. However, this particular study presented only a few snapshots (i.e., 28), and just one relatively small area in China (not the ones with rich ancient history like capitals or the areas along the rivers/Yellow River that nurtured Chinese agriculture). Why is this study so unique and important? This can be made clearer in the Introduction.

Response: Thank you for your helpful suggestion, and we apologize if this was unclear. We reorganized the introduction to make it more coherent, and also describe the novelty and the uniqueness of this study. Please see Line 25-74.

2. Also, please note that the current Introduction is quite similar to what's included in the Jia 2023 paper published at Regional Environmental Change, both the structure and argument of novelty. Quite a few sentences from the 2023 paper are used here again. This is not acceptable.

Response: Thank you for your helpful suggestion, and we apologize if this was confusing. We revised the structure and argument, and we also reorganized the introduction to make it more coherent. Please see Line 25-74.

3. Next, in terms of the methods used here compared with others published by the same group of authors including the 2023 one, any significant difference besides data/records used? Any improvement to the methods? Could we expect any improvement of methods from an additional paper? HYDE have already developed global scale LUC data, with even longer history and higher resolution, and this study has always compared their results with HYDE. From what angle can we justify that this dataset has "higher reliability" or can "improve the accuracy and reliability"? Comparing a regional study with global work, or filling a few missing data (aim 1) do not make this a better paper. The authors need to better clarify the intention, methods, and even the comparison in the discussion.

Response: Thank you for your insightful suggestion, and we apologize if this was confusing. Compare to 1000-1200, we developed cropland calculation indicators for 1300-1600 corresponding to different population categories (Please see Table 1, Line 592-614), and the algorithm to reconstruct the cropland by population is different (Please see the supplement material).

In this study, we used the improved historical cropland reconstruction methods to reconstruct 28 time-points cropland area by assimilating multiple data sources. Reconstruction of cropland area from 1000 to 1600 primarily relies on historical documents, population data. Furthermore, we used the most authoritative historical population data in China: "*History of Population of China*" and the cropland calculation indicators during this period corresponding to different population categories (Please see Table 1, Line 592-614 and the supplement material). We also attempt to analyze the

rationality of our dataset based on the population changes, settlements changes, warfare, and land policies that may have influenced land cultivation in Northeast China during the Liao, Jin, Yuan, and Ming periods (1000-1600) (Please Line 536-578, Fig. S1).

Overall, we reorganized supplemented some content attempt to better describe the intention, methods, and the comparison in the discussion. Please see Line 25-74, 211-261, 416-625 and the supplement material.

4. L24: again, the Introduction is quite similar to Jia 2023, this has to be revised to be acceptable anywhere?

Response: Thank you again for your helpful suggestion, and we apologize if this was confusing. We reorganized the introduction to make it more coherent. Please see Line 25-74.

5. L52: aims not aim.

Response: Thank you for your suggestion. We reorganized the aims to make it more coherent. Please see Line 72-74.

6. L55-57: how many aims do you have exactly? Two or four? These do not seem to be complete sentences.

Response: Thank you for your suggestion, and we apologize if this was confusing. We reorganized the aims to make it more coherent. Please see Line 70-74.

7. L113: this seem to be quite large for per person, can this value be used for the whole region?

Response: Thank you for your helpful suggestion, and we apologize if this was confusing. The definition of the Man is the adult labor force of a household (a male between the ages of 15 and 50 years in the Liao Dynasty; a male between the ages of 17 and 59 years in the Jin Dynasty; a male between the ages of 15 and 59 years in the Yuan Dynasty; a male between the ages of 16 and 60 years in the Ming Dynasty). The conclusion of 14 *Mu* per Man for agricultural population during this period (1000~1600)

is primarily derived from historical records in the Jin Dynasty (1200) and the relationship between population and cropland in the early Qing Dynasty (1661~1680) (Jia et al., 2023). And we also discussed the uncertainty of this value. Please see Line 115, 121, 143, 165, 592-614.

References:

Jia, R., Fang, X., and Ye, Y.: Gridded reconstruction of cropland cover changes in Northeast China from ad 1000 to 1200, Reg. Envir. Chang., 23, 128, https://doi.org/10.1007/s10113-023-02118-y, 2023.

8. L384: The method is done by now, but how did you compute the spatial distribution of cropland across time? The previous methods mainly focused on total area numbers, but should the spatial pattern change with time, as the factors influencing cropland distribution change? For the area records, would the administrative region boundary change over time, which affect the statistics? Fig. 5 is an example that may be impacted by boundary changes.

Response: Thank you for your insightful suggestion.

From 1000 to 1600, the provincial-level administrative districts were derived from the Historical Atlas of China (Tan, 1982a; Tan, 1982b), and the cropland area during this period was reconstruction primarily relies on the population data in different provincial-level administrative districts.

The unified administration boundaries may affect the correct records of cropland. When we unified administration boundaries from 1700 to 1980, we referred to similar studies and adopted the similar method (Wei et al., 2019). Moreover, we performed this operation first at the time points with data records. After obtained all the cropland area at the modern administrative divisions of all time points, we performed linear interpolation and polynomial curve fitting to obtain the cropland area at standard time points, which had relatively less impact on cropland records. Please see Line 203-261.

In addition, a study has indicated that the county-level administrative divisions are the most stable administrative division level in Chinese history (Zhao et al., 2024). Even

so, the cropland area of each county estimated by this method in this study is still uncertain, and we have further described the uncertainty in Uncertainty analysis section. Please see Line 619-625.

References:

Tan, Q.: The Historical Atlas of China, Volume 7, SinoMaps Press, Beijing, China, 1982a (in Chinese).

Tan, Q.: The Historical Atlas of China, Volume 6, SinoMaps Press, Beijing, China, 1982b (in Chinese).

Wei, X. W. X., Ye, Y. Y. Y., Zhang, Q. Z. Q., Li, B. L. B., and Wei, Z. W. Z.: Reconstruction of cropland change in North China plain area over the past 300 years., Global & Planetary Change, 60-70, 2019.

Zhao, Y., Wang, K., Zhao, B., and Wang, F.: Spatio-temporal process and pattern of the establishment of county-level administrative divisions in China in the past 2200 years, Acta Geographica Sinica, 79, 890-908, https://doi.org/10.11821/dlxb202404005, 2024.

9. L455: there are several comparisons here, how do you justify that your estimates are better than others? Or do you suggest that as long as you have more data records then it should be more accurate?

Response: Thank you for your helpful suggestion, and we apologize if this was confusing. Our dataset assimilated multiple data sources (e.g.: historical documents, population data, garrison reclamation data, revised published results, statistical data, land survey data and RS data) and based on the improved historical cropland reconstruction methods (e.g.: cropland calculation indicators for different historical periods corresponding to different population categories), and the trend of increase and decrease of cropland area consistent with historical facts.

In addition, we acknowledge that the current paper's reliability, accuracy, or uncertainties assessments are not yet sufficiently comprehensive. We have made every effort to supplement the relevant assessments and uncertainty analysis as much as possible. And we deleted some description of the spatiotemporal variation

characteristics of cropland area. Please see Line 382-412, 416-625.

10. L514: this is NOT "uncertainty analysis", there is no "analysis" at all. Just some random discussions.

Response: Thank you for your helpful suggestion, and we apologize if this was confusing. We have reorganized the structure of the "Uncertainty analysis section", and added more specific and detailed description to analyze uncertainty. Please see Line 579-625.

11. L533: don't you think the conclusion is a bit too long?

Response: Thank you for your helpful suggestion. We have reorganized the "Conclusion section". Please see Line 630-649.

---

## Author Comment (AC3)

**Response to referee comments**

We thank the two reviewers and the editor for the precious and constructive suggestions to improve our manuscript. We carefully revised our manuscript and addressed the comments of each of the two reviewers. Please find our point-by-point response below.

**Referee #1**

General comments. According to the World Meteorological Organization, 2023 is the hottest year on record. Therefore, it is of great significance to develop a long-term cropland dataset to explore the climatic effects of human land use. This study reconstructs millennial cropland for Northeast China. Topics fits the aims and scope of the ESSD. The following comments and suggestions should be considered for revisions.

Response: We sincerely thank the reviewers for valuing our work. Thank you for your insightful comments which have improved our work greatly. Please find our point-by-point response below.

1. --First, why only area estimation, and no spatial reconstruction? A 5' × 5' cropland dataset is developed for Northeast China from AD 1000 to 1200 by these authors (Gridded reconstruction of cropland cover changes in Northeast China from AD 1000 to 1200. https://link.springer.com/article/10.1007/s10113-023-02118-y). But in this study, only provincial-level or county-level cropland area is available. Why? Obviously, the datasets reconstructed in this study cannot be used by climate modelers. In addition, in terms of data from 1000 to 1200 years, is there any improvement in this paper compared to the paper mentioned above (Gridded reconstruction of cropland cover changes in Northeast China from AD 1000 to 1200)?

Response: Thank you for your helpful suggestion. In this study, 1000 to 1600 corresponds to historical provincial-level administrative districts, while 1700 to 2020 corresponds to modern county-level administrative districts.

    The primary reasons are as follows: First, compared to the regional existing historical LUCC gridded reconstruction results, the cropland area data of administrative

units is relatively more fundamental and reliable. Our research experiments indicate that there remains a certain degree of uncertainty in gridded reconstruction of cropland over long historical periods, even when using human factors supported by historical data in the allocation model (such as historical settlement points) (Jia et al., 2023). Long-term historical cropland gridded reconstructions need to consider using allocation methods that match the historical facts of different periods in local area. In theory, based on this dataset, researcher could even create datasets with a resolution of 1 meter or less. If climate modelers need to use gridded cropland datasets, they can easily convert our dataset into a customized grid spatial resolution dataset according to their required time range and main influencing factors (such as natural factors, human factors, etc.). For instance, some studies concluded that the HYDE dataset can be used as a map of agricultural potential and crop suitability, especially in periods before the advent of satellite imagery (Yu and Lu, 2018; Yu et al., 2021).

Second, for the 1700 to 2020 corresponds to modern county-level administrative districts, the average spatial scale of these counties in this dataset ranges between 0.5°×0.5° and 1°×1°. Theoretically, when readers use this dataset to convert it into the gridded dataset they need, the error can be controlled within between 0.5°×0.5° and 1°×1° even if they don't use natural or human factors to guide the allocation.

Third, this dataset can be applied in a wide range of scenarios (such as carbon emission and carbon neutrality, climate data construction, ecological footprint, and biological population assessment, etc.). The bilingual format and the administrative boundaries consistent with the current county-level administrative units in China also facilitate its use by scholars in the humanities and social sciences worldwide.

Realistically, compared to the primary data and reconstruction methods of Jia et al. (2023), this study directly used the results of the cropland area (1000-1200) of the above study. The main difference is that, this dataset provides provincial-level cropland area data for three time points (1000-1200) within the current administrative boundaries of Northeast China, consistent with the boundaries of the other 25 time points in this dataset. We are also very pleased to offer the reconstructed gridded cropland dataset in Northeast China from 1000 to 1200 as a reference solution for readers.

References:

Yu, Z. and Lu, C.: Historical cropland expansion and abandonment in the continental U.S. During 1850 to 2016, Glob. Ecol. Biogeogr., 27, 322-333, https://doi.org/10.1111/geb.12697, 2018.

Yu, Z., Jin, X., Miao, L., and Yang, X.: A historical reconstruction of cropland in china from 1900 to 2016, Earth Syst. Sci. Data, 13, 3203-3218, https://doi.org/10.5194/essd-13-3203-2021, 2021.

Jia, R., Fang, X., and Ye, Y.: Gridded reconstruction of cropland cover changes in Northeast China from ad 1000 to 1200, Reg. Envir. Chang., 23, 128, https://doi.org/10.1007/s10113-023-02118-y, 2023.

2. --Second, the applicability of the reconstruction method of estimating the cropland area for a small area by population. Generally speaking, estimating cropland by population is mostly applicable at continental to global scales. In the case of a small region, more other factors will affect the relationship between population and cropland. Response: Thank you for your suggestion. The fundamental reason is that there are no direct historical records of cropland area in Northeast China from 1000 to 1600. When we selecting proxy indicators to reconstruct cropland area, the population data for this period is relatively complete and authoritative.

We acknowledge that at a regional scale, the factors affecting the relationship between population and cropland area are diverse and may change over time. Therefore, we combine historical facts (particularly from 1000 to 1600) and use the most authoritative historical population data in China: "*The History of Population of China*" (Wu and Ge, 2005a; Cao and Ge, 2005b). Based on the needs of this study, we extracted the population data (mainly divided into agricultural, non-agricultural, and military populations) that was consistent with the scope of this study area, and developed cropland calculation indicators for different historical periods corresponding to different population categories.

In several global LUCC datasets, such as HYDE, when estimating cropland based

on population, either the population indicators (e.g.: per capita cropland) remain unchanged across different historical periods; or in the uncertainty estimates, homogeneously varied it with time across the globe; or to account for their uncertainties, different population databases were used and the upper and lower ends of an uncertainty range were assessed. However, these estimation methods can still lead to errors in cropland area on a global or regional scale. Therefore, when this study used historical population data to reconstruct cropland, special attention was paid to two key points: 1) authoritative and accurate population data, and 2) cropland areas corresponding to different population categories based on historical facts (Please see Line 632-654). Thus, we believe this method is relatively applicable for Northeast China from 1000 to 1600.

***Line 632-654***: (3) The two proxy indicators of 14 *Mu* (0.93 hm$^2$, the average annual potential cropland area per Man of the agricultural population) and 2 *Mu* (0.13 hm$^2$, the average cropland area per household in the nonagricultural population) from 1000 to 1600 may lead to inaccuracies in cropland estimation. The reasons for using population to reconstruct cropland during this period have been detailed in the previous section, necessitating further analysis and clarification of the corresponding cropland-related indicators.

Firstly, the conclusion of 14 *Mu* per Man for agricultural population during the Liao and Jin Dynasties (1000~1200) is primarily derived from historical records in the Jin Dynasty (1200) and the relationship between population and cropland in the early Qing Dynasty (1661~1680) (Jia et al., 2023). There are two reasons why 14 *Mu* was used in the Yuan and Ming Dynasties (1300~1600): one reason is the agricultural household size and the ratios of Man in agricultural household in Northeast China during the Yuan and Ming Dynasties (1300~1600) are closer to those of the Liao and Jin Dynasties (1000~1200) (Table 1). And the per capita cropland area owned by agricultural population in the Liao-Jin-Yuan-Ming periods (1000~1600) consistently ranged between 4 and 5 *Mu* (0.27~0.33 hm$^2$), slightly higher than the subsistence level of 3 *Mu* per capita in previous studies for the same historical period in this region (Ye et al., 2009; Fang et al., 2006; Shi, 1990), which is relatively reasonable. The second reason is that there were no significant changes in agricultural production technology in Northeast China during the Liao-Jin-Yuan-Ming periods (1000~1600), and the population declined significantly compared with the Liao and Jin Dynasties (1000~1200) due to factors such as warfare. However, considering the social stability at standard time-points during the Yuan and Ming Dynasties (1300~1600), the strong willingness of the

agricultural population towards cultivation, and the limitations of individual cultivation capabilities, the cropland from the Liao and Jin Dynasties could be relatively easily inherited and reclaimed by descendants.

Secondly, Similar to the agricultural population, considering the non-agricultural household size, stable agricultural production technology, the historical inheritance of most ethnic groups, this study continues to use 2 *Mu* as the calculation indicator of non-agricultural population in the Yuan and Ming Dynasties (1300~1600) (Cong, 1993a; Cong, 1993b; Wu and Ge, 2005a; Cao and Ge, 2005b; Liu et al., 2016).

Response: Thank you for your insightful suggestion. We acknowledge that the current paper's reliability, accuracy, or uncertainties assessments are not yet sufficiently comprehensive. We have made every effort to supplement the relevant assessments and uncertainty analysis as much as possible. Please see Line 428-495; 619-678, new Fig. 6, new Fig. 7 and Table S3.

[revised manuscript text omitted]

The abbreviations used in the table are as follows: HYDE3.2 refers to Goldewijk et al. (2017); SAGE refers to Ramankutty et al. (2008) and Ramankutty and Foley. (1999); KK10 refers to Kaplan et al. (2011); PJ refers to Pongratz et al. (2008); Yu 2021 refers to Yu et al. (2021); CSY denotes the Chinese Statistical Yearbook (refer to provincial and prefectural statistical yearbook); NLS denotes the National Land Survey (1985 refer to the first general land investigation (Committee of Integrative Survey of Natural Resources and Committee of National Planning of Chinese Academy of Sciences, 1989); 1995 refers to

the first national land survey (Li, 2000); 2010 and 2015 refer to the second national land survey; 2020 refers to the third national land survey (https://gtdc.mnr.gov.cn/Share#/)); Zhang 1991 refers to Zhang (1991); Zhou 2001 refers to Zhou (2001); Shi 2015 refers to Shi (2015); Jin 2015 refers to Jin et al. (2015); Li 2016 refers to Li et al. (2016); He 2023 refers to He et al. (2023).

4. --Fourth, writing is not done from the perspective of data development (Data description paper), it is more like a research paper. For example, the core content of the results should not be the analysis of the spatio-temporal characteristics of cropland changes, but the rationality, reliability, accuracy, and potential uses of the data products developed in this paper. More specific comments are as follows.

Response: Thank you for your insightful suggestion. We reorganized the paper to make it more like a data description paper. It mainly includes: 1. Deletion of the conclusion section (Please see "Conclusion section", Line 684-703); 2. Reorganized of introduction, results and discussion sections (Please see "Introduction section", Line 25-82; "Results section", Line 393-426; "Discussion section", Line 427-678); 3. Add the credibility assessment and uncertainty analysis (Please see Line 428-495; 619-678); 4. Potential uses are added to the introduction and abstract sections (Please see Line 22-24, 39-41).

*Line 22-24*: Additionally, the dataset shows relatively better credibility assessment results, which can provide a refined data base for historical LUCC dataset reconstruction, carbon emission estimation, climate data construction, etc. The dataset can be downloaded from https://doi.org/10.6084/m9.figshare.25450468.v2 (Jia, 2024).

*Line 39-41*: have been extensively employed in global change research. Such as carbon emission and carbon neutrality (Xu et al., 2024), climate data construction (Gortan et al., 2024), ecological footprint (Wang et al., 2024), and biological population assessment (Ye et al., 2024), etc.

5. --Title and Introduction. Why reconstruction for 1000 to 2020 in Northeast China? More explanations are necessary. Based on Figure 5, From 1000-1700, there was only a small area of cropland in Northeast China. Line 393, In 1200, cropland fraction of 1.17%; In 1400, line 395, cropland fraction of only 0.19%. The environmental impact

30

of such a small area of cropland is completely negligible. Based on figure 5, the topic for past 300 years (Ye, Y., Fang, X., Ren, Y., Zhang, X., and Chen, L.: Cropland cover change in northeast china during the past 300 years, Science China Earth Sciences, 52, 1172-1182, https://doi.org/10.1007/s11430-009-0118-8, 2009.) is good, but for 1000 to 2020 may be not a good research topic.

Response: Thank you for your insightful suggestion. We chose to reconstruct the cropland changes in Northeast China over the past millennium mainly for the following reasons: 1. From the LUCC projects carried out by IGBP and IHDP in the last century to the recent PAGES (The PAGES (Past Global Changes) project is an international effort to coordinate and promote past global change research. The primary objective is to improve our understanding of past changes in the Earth system in order to improve projections of future climate and environment, and inform strategies for sustainability.), LandCover6k (The goal of LandCover6k was to produce datasets on past land-cover and land-use on continental and global spatial scales that are useful for climate modeling studies on land-use as a climate forcing.) and other research projects, the importance of accurate long-term historical LUCC datasets has been emphasized.

2. The land reclamation in Northeast China exhibited a unique pattern: during the period between the two land reclamations (eleventh and twelfth centuries; from the nineteenth century to present), there was a prolonged period of nomadism in this area (Jia et al., 2023). In addition, the global historical LUCC datasets fail to demonstrate the historical fact of cropland cultivation in the study area from 1000 to 1200.

3. We want to better demonstrate the changes in human impact on terrestrial environments, ranging from near-natural original states (natural dominance) to significant alterations induced by extensive human intervention (anthropogenic dominance), particularly since the Industrial Revolution.

4. Northeast China has now become one of the most important agricultural regions in China and the world. Focusing solely on the past 300 years of research is not conductive to the exploration of long-term LUCC effects in critical agricultural areas worldwide (He et al., 2023). If feasible, we are willing to expand the dataset regarded as "truth values" to encompass a broader historical span, thereby enhancing the

credibility of historical cropland area.

5. We also reorganized the introduction to describe the novelty of this manuscript (Please see Line 73-82).

*Line 73-82*: The dataset in this study presents a critical update and extension of the former historical cropland cover change in the three provinces of Northeast China over the past 300 years (Ye et al., 2009). Throughout the prolonged agricultural development, the natural vegetation landscape in the Northeast China has undergone notable transformations. In this study, we used the improved historical cropland reconstruction methods to reconstruct 28 time-points cropland area by assimilating multiple data sources in Northeast China from 1000 to 2020. The mainly new features of this dataset include: (1) Extended the reconstruction period to 1000~2020, aligning with the standard time-points of internationally established global historical LUCC datasets; (2) the reconstruction included the entire East of Inner Mongolia, which area accounts for approximately 45% of the Northeast China. (3) the smallest administrative divisions for the reconstructed cropland are at the provincial-level from 1000 to 1600, and at the county-level from 1700 to 2020. Our main objective is to provide a long-term time series of cropland area change dataset in Northeast China that is close to the historical "truth value" under a unified standard.

historical land use reconstruction, estimating cropland area based on population as a proxy is only applicable to large-scale scales such as global and continental. For example, HYDE uses population to estimate the world's historical cropland. It also makes sense to reconstruct China's historical cropland in this way. But the Northeast is only a small region of China, so there is a lot of uncertainty in the results of this estimate.

Response: Thank you for your constructive suggestion, and we apologize if this was confusing. The main algorithm applied in the Liao, Jin, Yuan, and Ming Dynasties (1000-1600) can be found in the supplementary materials.

The reason and the credibility of using population to reconstruct cropland is similar to your second question, to which we have already responded and discussed more fully in the "Uncertainty analysis section". Please see Line 632-654. Regarding the importance of Northeast China, we added content in the "Introduction section". Please see Line 60-72.

*Line 632-654*: **4.4 Uncertainty analysis**

[revised manuscript text omitted]

7. --In addition, for 1000, 1100, and 1200, what's the difference between this study and

the paper mentioned above (Gridded reconstruction of cropland cover changes in Northeast China from AD 1000 to 1200).

Response: Thank you for your comment. This study is a continuation of a previous study. Realistically, compared to the primary data and reconstruction methods of Jia et al. (2023), this study directly used the results of the cropland area (1000-1200) of the above study. The dataset in this study presents a critical update and extension of the former historical cropland cover change in the three provinces of Northeast China over the past 300 years (Ye et al., 2009). The main difference is that, this dataset provides provincial-level cropland area data for three time points (1000-1200) within the current administrative boundaries of Northeast China, consistent with the boundaries of the other 25 time points in this dataset. We are also very pleased to offer the reconstructed gridded cropland dataset in Northeast China from 1000 to 1200 as a reference solution for readers.

[revised manuscript text omitted]

| Counties where cropland need to be supplemented | Year | Adjacent counties for interpolation or based on the number of settlements |
|---|---|---|
| Nong'an County | | Revised according to local gazetteer of Nong'an County |
| Yongji County | | Based on the historical origin of this region and Jilin City, revised according to the average proportional relationship between the two regions in 1908, 1914, 1931 and 1940 |
| Huadian City | | Dunhua City, Jingyu County, Fusong County |
| Shulan City | | Jilin City, Changchun City, Jiaohe City, Wuchang City |
| Panshi City | 1800, 1850 | Huadian City, Dongliao County, Dongfeng County, Meihekou City, Huinan County |
| Mulan County | | Bayan County, Tonghe County, Bin County |
| Fuyu County | | Yi'an County, Nehe City, Lindian County, Qiqihar City, Gannan County |
| Bei'an City | | Kedong County, Wudalianchi City, Hailun City |
| Jiaohe City | | Interpolated based on multiplying the number of |

38

| | | |
|---|---|---|
| Changling County | | settlements and the cropland area owned by unit settlement in this region in 1800 and 1850 respectively |
| Hunchun City | | |
| Dunhua City | | |
| Siping City | | |
| Lishu County | | |
| Yitong Manchu Autonomous County | | |
| Gongzhuling City | | |
| Shuangliao City | | |
| Yanji City | | Revised according to local gazetteer of Yanji City |
| Tonghe County | 1850 | Yilan County, Mulan County, Fangzheng County |
| Lanxi County | | Suihua City, Zhaodong City, Harbin City, Qinggang County |

10. --Results. ESSD readers are more concerned about the reliability, availability, and accuracy of data products. However, the spatiotemporal variation characteristics of cropland area are not the most important.

Response: Thank you for your helpful suggestion. We acknowledge that the current paper's reliability, accuracy, or uncertainties assessments are not yet sufficiently comprehensive. We have made every effort to supplement the relevant assessments and uncertainty analysis as much as possible. And we deleted some description of the spatiotemporal variation characteristics of cropland area. Please see "Results section", Line 393-426; "Discussion section", Line 427-678.

11. --4.1 comparison. The comparison with the global dataset does not indicate the reliability of the dataset developed in this paper, because the global dataset itself has a large degree of uncertainty. The fact that the reconstruction results in this paper are very different from the global dataset does not mean that the dataset developed in this paper is reliable. Line 516-517, the following statement is not acceptable "Comparative analysis with global historical LUCC datasets indicates that the results of this study are relatively credible and more rational."

Response: Thank you for your insightful suggestion. "Comparative analysis with global historical LUCC datasets indicates that the results of this study are relatively credible and more rational." This statement is indeed inappropriate. We have deleted such

statements in the original text and used more objective language to describe the differences between different datasets and analyze the possible reasons for the differences. Please see "Discussion section", Line 427-678.

12. -- technical corrections. Figure 5, no titles for x and y axes.

Response: Thank you for your suggestion. We revised the Fig. 4 and Fig. 7 in current manuscript.

[Figure]

**Figure 4: Changes in total cropland area in the Northeast China from 1000 to 2020.**

[Figure]

**Figure 7: Comparison of total cropland area from global historical LUCC datasets, previous studies and this study in the Northeast China.**

Once again, thank you very much for your constructive comments, which has greatly improved this manuscript.

---

## Author Comment (AC4)

**Response to referee comments**

We thank the two reviewers and the editor for the precious and constructive suggestions to improve our manuscript. We carefully revised our manuscript and addressed the comments of each of the two reviewers. Please find our point-by-point response below.

**Referee #2**

The authors developed a dataset to document cropland area over the past 1000 years in the North China. By using historical records and recent datasets, the manuscript particularly looked at the spatial changes and possible improvement to the accuracy of the regional dataset. I have a few concerns and suggestions for the authors to consider if they decide to revise the manuscript.

Response: Thank you very much for your constructive comments on our manuscript. We appreciate the time you spent reviewing our manuscript. Please find our point-by-point response below.

1. The novelty of this manuscript is not clearly presented. The authors have already published a few similar papers in the past few years, and even one for the Northeast China region. The only difference is the time period covered here. Land use change, especially for such long history with spatial coverage, is deemed important in understanding carbon budget, land emissions, and many other studies. This is what the authors also emphasized in the Introduction. However, this particular study presented only a few snapshots (i.e., 28), and just one relatively small area in China (not the ones with rich ancient history like capitals or the areas along the rivers/Yellow River that nurtured Chinese agriculture). Why is this study so unique and important? This can be made clearer in the Introduction.

Response: Thank you for your helpful suggestion, and we apologize if this was unclear. We reorganized the introduction to make it more coherent (Please see "Introduction section", Line 25-82.), and also describe the novelty and the uniqueness of this study. Please see Line 73-82.

*Line 73-82*: The dataset in this study presents a critical update and extension of the former historical cropland cover change in the three provinces of Northeast China over the past 300 years (Ye et al., 2009). Throughout the prolonged agricultural development, the natural vegetation landscape in the Northeast China has undergone notable transformations. In this study, we used the improved historical cropland reconstruction methods to reconstruct 28 time-points cropland area by assimilating multiple data sources in Northeast China from 1000 to 2020. The mainly new features of this dataset include: (1) Extended the reconstruction period to 1000~2020, aligning with the standard time-points of internationally established global historical LUCC datasets; (2) the reconstruction included the entire East of Inner Mongolia, which area accounts for approximately 45% of the Northeast China. (3) the smallest administrative divisions for the reconstructed cropland are at the provincial-level from 1000 to 1600, and at the county-level from 1700 to 2020. Our main objective is to provide a long-term time series of cropland area change dataset in Northeast China that is close to the historical "truth value" under a unified standard.

2. Also, please note that the current Introduction is quite similar to what's included in the Jia 2023 paper published at Regional Environmental Change, both the structure and argument of novelty. Quite a few sentences from the 2023 paper are used here again. This is not acceptable.

Response: Thank you for your helpful suggestion, and we apologize if this was confusing. We revised the structure and argument, and we also reorganized the introduction to make it more coherent. Please see "Introduction section", Line 25-82.

3. Next, in terms of the methods used here compared with others published by the same group of authors including the 2023 one, any significant difference besides data/records used? Any improvement to the methods? Could we expect any improvement of methods from an additional paper? HYDE have already developed global scale LUC data, with even longer history and higher resolution, and this study has always compared their results with HYDE. From what angle can we justify that this dataset has "higher reliability" or can "improve the accuracy and reliability"? Comparing a regional study with global work, or filling a few missing data (aim 1) do not make this a better paper. The authors need to better clarify the intention, methods, and even the comparison in

the discussion.

Response: Thank you for your insightful suggestion, and we apologize if this was confusing. Compare to 1000-1200, we developed cropland calculation indicators for 1300-1600 corresponding to different population categories (Please see Table 1, Line 632-654), and the algorithm to reconstruct the cropland by population is different (Please see the supplement material).

**Table 1: The index of cropland area reconstruction from 1000 to 1600**

[revised manuscript text omitted]

Overall, we reorganized supplemented some content attempt to better describe the intention (Please see "Introduction section", Line 25-82), methods (Line 219-269), and the comparison in the discussion (Please see "Discussion section", Line 427-678) and the supplement material.

*Line 219-269*: (3) Linear interpolation and polynomial curve fitting to obtain the cropland area: Previous

studies have used the linear interpolation and polynomial curve fitting to reconstruct cropland areas (He et al., 2017; Jin et al., 2015; Ramankutty and Foley, 1999; Wei et al., 2016; Wei et al., 2021; Ye et al., 2015; Yu, 2019; Fang et al., 2021), and the interpolated data did not reduce the credibility of their datasets. In addition, previous studies have shown that in the process of reclamation in the Northeast China over the past 300 years, 1860 was a dividing point between slow growth and rapid growth, mainly due to the implementation of the immigration and reclamation policy by the Qing government (Fang et al., 2020; Ye et al., 2009; Fang et al., 2005; Kong and Feng, 1989). Therefore, this study selected the CNEC data (Ye et al., 2009) in 1683, 1735, 1780, 1908 and 1914 for linear interpolation and polynomial curve fitting of cropland area data for each county or district in the three provinces of the Northeast China, obtaining data for 1700, 1750, 1800, 1850 and 1900. In addition, this study selected the data from Tian (Tian, 2005) in 1724, 1782, 1868, and 1911; the CNEC data (Ye et al., 2009) in 1735; the data from Ye (Ye and Fang, 2012) in 1916 for linear interpolation and polynomial curve fitting to obtain cropland area data for 1700, 1750, 1800, 1850, and 1900 in the Eastern of Inner Mongolia. The problems that may be encountered during the operation and the corresponding solutions are as follows:

[revised manuscript text omitted]

in Table S2.

4. L24: again, the Introduction is quite similar to Jia 2023, this has to be revised to be acceptable anywhere?

Response: Thank you again for your helpful suggestion, and we apologize if this was confusing. We reorganized the introduction to make it more coherent. Please see "Introduction section", Line 25-82.

5. L52: aims not aim.

Response: Thank you for your suggestion. We reorganized the aims to make it more coherent. Please see Line 81-82.

***Line 81-82***: Our main objective is to provide a long-term time series of cropland area change dataset in Northeast China that is close to the historical "truth value" under a unified standard.

6. L55-57: how many aims do you have exactly? Two or four? These do not seem to be complete sentences.

Response: Thank you for your suggestion, and we apologize if this was confusing. We reorganized the aims to make it more coherent. Please see Line 73-82.

***Line 73-82***: The dataset in this study presents a critical update and extension of the former historical cropland cover change in the three provinces of Northeast China over the past 300 years (Ye et al., 2009). Throughout the prolonged agricultural development, the natural vegetation landscape in the Northeast China has undergone notable transformations. In this study, we used the improved historical cropland reconstruction methods to reconstruct 28 time-points cropland area by assimilating multiple data sources in Northeast China from 1000 to 2020. The mainly new features of this dataset include: (1) Extended the reconstruction period to 1000~2020, aligning with the standard time-points of internationally established global historical LUCC datasets; (2) the reconstruction included the entire East of Inner Mongolia, which area accounts for approximately 45% of the Northeast China. (3) the smallest administrative divisions for the reconstructed cropland are at the provincial-level from 1000 to 1600, and at the county-level from 1700 to 2020. Our main objective is to provide a long-term time series of cropland area change dataset in Northeast China that is close to the historical "truth value" under a unified standard.

7. L113: this seem to be quite large for per person, can this value be used for the whole region?

Response: Thank you for your helpful suggestion, and we apologize if this was confusing. The definition of the Man is the adult labor force of a household (a male between the ages of 15 and 50 years in the Liao Dynasty; a male between the ages of 17 and 59 years in the Jin Dynasty; a male between the ages of 15 and 59 years in the Yuan Dynasty; a male between the ages of 16 and 60 years in the Ming Dynasty). The conclusion of 14 *Mu* per Man for agricultural population during this period (1000~1600) is primarily derived from historical records in the Jin Dynasty (1200) and the relationship between population and cropland in the early Qing Dynasty (1661~1680) (Jia et al., 2023). And we also discussed the uncertainty of this value. Please see Line 123 (a male between the ages of 15 and 50 years in the Liao Dynasty), 129 (a male between the ages of 17 and 59 years in the Jin Dynasty), 151 (a male between the ages of 15 and 59 years in the Yuan Dynasty), 172-173 (a male between the ages of 16 and 60 years in the Ming Dynasty), 632-654.

*Line 632-654*: (3) The two proxy indicators of 14 *Mu* (0.93 hm$^2$, the average annual potential cropland area per Man of the agricultural population) and 2 *Mu* (0.13 hm$^2$, the average cropland area per household in the nonagricultural population) from 1000 to 1600 may lead to inaccuracies in cropland estimation. The reasons for using population to reconstruct cropland during this period have been detailed in the previous section, necessitating further analysis and clarification of the corresponding cropland-related indicators.

Firstly, the conclusion of 14 *Mu* per Man for agricultural population during the Liao and Jin Dynasties (1000~1200) is primarily derived from historical records in the Jin Dynasty (1200) and the relationship between population and cropland in the early Qing Dynasty (1661~1680) (Jia et al., 2023). There are two reasons why 14 *Mu* was used in the Yuan and Ming Dynasties (1300~1600): one reason is the agricultural household size and the ratios of Man in agricultural household in Northeast China during the Yuan and Ming Dynasties (1300~1600) are closer to those of the Liao and Jin Dynasties (1000~1200) (Table 1). And the per capita cropland area owned by agricultural population in the Liao-Jin-Yuan-Ming periods (1000~1600) consistently ranged between 4 and 5 *Mu* (0.27~0.33 hm$^2$), slightly higher than the

subsistence level of 3 *Mu* per capita in previous studies for the same historical period in this region (Ye et al., 2009; Fang et al., 2006; Shi, 1990), which is relatively reasonable. The second reason is that there were no significant changes in agricultural production technology in Northeast China during the Liao-Jin-Yuan-Ming periods (1000~1600), and the population declined significantly compared with the Liao and Jin Dynasties (1000~1200) due to factors such as warfare. However, considering the social stability at standard time-points during the Yuan and Ming Dynasties (1300~1600), the strong willingness of the agricultural population towards cultivation, and the limitations of individual cultivation capabilities, the cropland from the Liao and Jin Dynasties could be relatively easily inherited and reclaimed by descendants.

Secondly, Similar to the agricultural population, considering the non-agricultural household size, stable agricultural production technology, the historical inheritance of most ethnic groups, this study continues to use 2 *Mu* as the calculation indicator of non-agricultural population in the Yuan and Ming Dynasties (1300~1600) (Cong, 1993a; Cong, 1993b; Wu and Ge, 2005a; Cao and Ge, 2005b; Liu et al., 2016).

Response: Thank you for your helpful suggestion, and we apologize if this was confusing. Our dataset assimilated multiple data sources (e.g.: historical documents, population data, garrison reclamation data, revised published results, statistical data, land survey data and RS data) and based on the improved historical cropland reconstruction methods (e.g.: cropland calculation indicators for different historical periods corresponding to different population categories), and the trend of increase and decrease of cropland area consistent with historical facts.

In addition, we acknowledge that the current paper's reliability, accuracy, or uncertainties assessments are not yet sufficiently comprehensive. We have made every effort to supplement the relevant assessments and uncertainty analysis as much as possible. And we deleted some description of the spatiotemporal variation

[revised manuscript text omitted]

The abbreviations used in the table are as follows: HYDE3.2 refers to Goldewijk et al. (2017); SAGE refers to Ramankutty et al. (2008) and Ramankutty and Foley. (1999); KK10 refers to Kaplan et al. (2011); PJ refers to Pongratz et al. (2008); Yu 2021 refers to Yu et al. (2021); CSY denotes the Chinese Statistical Yearbook (refer to provincial and prefectural statistical yearbook); NLS denotes the National Land Survey (1985 refer to the first general land investigation (Committee of Integrative Survey of Natural Resources and Committee of National Planning of Chinese Academy of Sciences, 1989); 1995 refers to

the first national land survey (Li, 2000); 2010 and 2015 refer to the second national land survey; 2020 refers to the third national land survey (https://gtdc.mnr.gov.cn/Share#/)); Zhang 1991 refers to Zhang (1991); Zhou 2001 refers to Zhou (2001); Shi 2015 refers to Shi (2015); Jin 2015 refers to Jin et al. (2015); Li 2016 refers to Li et al. (2016); He 2023 refers to He et al. (2023).

10. L514: this is NOT "uncertainty analysis", there is no "analysis" at all. Just some random discussions.

Response: Thank you for your helpful suggestion, and we apologize if this was confusing. We have reorganized the structure of the "Uncertainty analysis section", and added more specific and detailed description to analyze uncertainty. Please see "Uncertainty analysis section", Line 619-678.

11. L533: don't you think the conclusion is a bit too long?

Response: Thank you for your helpful suggestion. We have reorganized the "Conclusion section". Please see "Conclusion section", Line 684-703.

Once again, thank you very much for your constructive comments, which has greatly improved this manuscript.